# Compound marine heatwaves and ocean acidity extremes

**Friedrich A. Burger** [1,2] ✉, **Jens Terhaar** [1,2] & **Thomas L. Frölicher** [1,2]

Compound MHW-OAX events, during which marine heatwaves (MHWs) co-occur with ocean acidity extreme (OAX) events, can have larger impacts on marine ecosystems than the individual extremes. Using monthly open-ocean observations over the period 1982–2019, we show that globally 1.8 in 100 months (or about one out of five present-day MHW months) are compound MHW-OAX event months under a present-day baseline, almost twice as many as expected for 90th percentile extreme event exceedances if MHWs and OAX events were statistically independent. Compound MHW-OAX events are most likely in the subtropics (2.7 in 100 months; 10°–40° latitude) and less likely in the equatorial Pacific and the mid-to-high latitudes (0.7 in 100 months; >40° latitude). The likelihood pattern results from opposing effects of temperature and dissolved inorganic carbon on [H⁺]. The likelihood is higher where the positive effect on [H⁺] from increased temperatures during MHWs outweighs the negative effect on [H⁺] from co-occurring decreases in dissolved inorganic carbon. Daily model output from a large-ensemble simulation of an Earth system model is analyzed to assess changes in the MHW-OAX likelihood under climate change. The projected long-term mean warming and acidification trends have the largest effect on the number of MHW-OAX days per year, increasing it from 12 to 265 days per year at 2 °C global warming relative to a fixed pre-industrial baseline. Even when long-term trends are removed, an increase in [H⁺] variability leads to a 60% increase in the number of MHW-OAX days under 2 °C global warming. These projected increases may cause severe impacts on marine ecosystems.

Anthropogenic climate change has led to an increase in frequency and intensity of ocean extreme events, such as marine heatwaves[1–7] and ocean acidity extremes[8,9], a trend that is projected to continue over the 21st century[3,9]. The predominantly harmful impacts of such individual extreme events on marine ecosystems[10–12] may become more severe when two extreme events occur together[13–15]. These so-called compound events[16,17] have been vastly studied on land[18,19], whereas marine compound events, such as compound MHW-OAX events (events of unusually high temperature and hydrogen ion concentration, [H⁺]), are just starting to receive more attention[20,21].

The impact of compound MHW-OAX events on marine biota may exceed the impact from individual MHW or OAX events since co-occurring extremes can interact synergistically[22]. For example, the combination of high temperature and acidity conditions negatively impacted pteropods across cellular, physiological, and population levels in the California Current System in 2016[13,15], and some of the devastating impacts of the Northeast Pacific 2013–2015 MHW[23] may have been amplified by the co-occurring extreme OA conditions[21]. Such rare observations and modeling studies of MHW-OAX events in the ocean are further corroborated by laboratory experiments

[1]Climate and Environmental Physics, Physics Institute, University of Bern, Bern, Switzerland. [2]Oeschger Centre for Climate Change Research, University of Bern, Bern, Switzerland. ✉e-mail: friedrich.burger@unibe.ch

showing synergistic negative effects on calcification, reproduction, and survival[24], and trends towards lower survival, growth, and development[25] for many species under compound MHW-OAX conditions compared to single extreme conditions, and by mesocosm experiments showing shifts in community structure[26,27].

The likelihood of compound MHW-OAX event occurrence is influenced by a complex interplay of direct and indirect effects of high temperatures on [H$^+$] during MHWs. While hot temperatures directly lead to increases in [H$^+$] via changes in the carbonate chemistry equilibrium[28], they can also modulate [H$^+$] indirectly via changes in dissolved inorganic carbon (C$_T$)[21]. These indirect changes of [H$^+$] during MHWs include a reduction of the CO$_2$ solubility in surface waters[29] and an associated net release of oceanic CO$_2$ to the atmosphere that reduces C$_T$ and [H$^+$][30], an increase in upper ocean thermal stratification[5] resulting in suppressed mixing of surface waters with carbon-rich subsurface waters and hence a reduction in surface ocean C$_T$ and [H$^+$][31], as well as changes in organic matter production that reduce C$_T$ and [H$^+$] if production increases[31]. The knowledge on compound MHW-OAX events and their drivers is currently very limited due to a lack of direct observations. However, novel observational-based data products[9,32,33] and large-ensemble simulations of comprehensive Earth System Models (ESMs)[34,35] permit us now to study compound MHW-OAX events globally.

Here, we characterize patterns, identify drivers, and assess future changes of compound MHW-OAX events using (i) global monthly gridded observation-based sea surface temperature (SST)[33] and surface [H$^+$] data[9] from 1982 to 2019, (ii) time-series data of temperature and [H$^+$] from fifteen ocean stations with either approximately monthly or 3-hourly measurement frequency collected between 1983 and 2020, and (iii) output from 30 ensemble members of the Earth System Model GFDL ESM2M[36,37] at daily-mean resolution covering the period from 1861 to 2100 (see "Methods"). The analysis is restricted to the open ocean, since the high variability and locally important processes in coastal oceans, such as riverine fluxes or shelf and coastal dynamics, are neither captured by the gridded observation-based product[32,38] nor by the GFDL ESM2M model[37]. Extreme events in SST and [H$^+$] are defined with respect to seasonally varying 90th percentiles (see Methods). The percentile thresholds are determined from the shortest available timestep, i.e., monthly for observations and daily for the model output. In this study, compound MHW and OAX events are multivariate compound events[19] during which both extreme hot temperature and high acidity conditions co-occur in space and time. We refer to them simply as "compound MHW-OAX events". We here quantify the number of MHW-OAX months per year (for the observation-based data) and the number of MHW-OAX days per year (daily-mean model output), irrespective of whether these months or days belong to the same ongoing MHW-OAX event. Similarly, we define the likelihood multiplication factor (LMF) that quantifies the likelihood that a month or day is under MHW-OAX conditions relative to the expected likelihood if MHWs and OAX events would occur independently from each other. The LMF is the ratio between the observed likelihood of compound event months or days *p(MHW-OAX day or month)*, calculated as the percentage of months or days that are under MHW-OAX conditions over a given period, and the theoretical likelihood of compound event months or days if SST and [H$^+$] were statistically independent. The theoretical likelihood of compound event months or days for statistically independent variables is given by the product of the individual likelihoods of MHWs and OAX months or days *p(MHW day or month) × p(OAX day or month)*:[39]

$$\mathrm{LMF} = \frac{p(MHW\text{-}OAX\ day\ or\ month)}{p(MHW\ day\ or\ month) \times p(OAX\ day\ or\ month)} \quad (1)$$

For our definition of single extreme-event months or days (90th percentile thresholds), this theoretical likelihood is 10% • 10% = 1%,

corresponding to 0.12 months per year for monthly data or 3.65 days per year for daily model output under MHW-OAX conditions. An LMF higher than 1 indicates that compound event months or days occur more often than by chance and that more than 1% of all months or days are under MHW-OAX conditions. An LMF lower than one indicates a reduced likelihood of compound event months or days. As an example, if four out of 200 months were under MHW-OAX conditions, the LMF would be 2, meaning that the likelihood of a compound event month would be twice as large as under independence. However, if only one out of 200 months was under MHW-OAX conditions, the LMF would be 0.5, meaning that the likelihood of a compound event month would be only half as large as under independence.

## Results

### Present-day pattern of compound MHW-OAX event occurrence

The global gridded observation-based data shows that globally 1.8 in 100 months are compound MHW-OAX events (Fig. 1). This is 1.8 times more often (LMF = 1.8) than expected if variations in SST and [H$^+$] anomalies were statistically independent. The LMF is larger than one over 65% of the ocean surface area. Compound MHW-OAX extremes are most frequent in the subtropical regions (2.7 in 100 months or LMF = 2.7 over 40°–10° latitude) and least frequent in the equatorial Pacific (0.8 in 100 months or LMF = 0.8 over 10°S–10°N latitude) and the high latitudes (0.7 in 100 months or LMF = 0.7 over 40°–80° latitude; Fig. 1). These regions are separated by the contour of LMF equal to one (thin gray contour line in Fig. 1) that follows closely the subpolar fronts in both hemispheres and the El Niño/Southern Oscillation region in the central and eastern tropical Pacific.

### Potential drivers of MHW-OAX events

To better understand the regional differences in the occurrence of MHW-OAX events and possible future changes, we now quantify the underlying drivers and discuss physical and biogeochemical processes. Mathematically, the LMF of MHW-OAX events cannot be decomposed into contributions from its drivers. However, the LMF can be estimated from the Pearson correlation coefficient (in the following simply correlation coefficient) of SST and [H$^+$] anomalies (Supplementary Fig. 1; see "Methods"), which can be mathematically decomposed. The contributions to the correlation coefficient (Fig. 2a) include the direct contribution from variations in SST (Fig. 2b), the contribution from variations in salinity-normalized dissolved inorganic carbon (sC$_T$; Fig. 2c), as well as smaller contributions from variations in salinity-normalized alkalinity (sA$_T$) and a freshwater cycling term (see "Methods" and Supplementary Fig. 2). The freshwater cycling term quantifies the direct impact from salinity variations (through changes in precipitation or evaporation, or changes in ocean circulation) and the changes in C$_T$ and A$_T$ that are proportional to these salinity variations. Globally, the SST contribution increases the correlation coefficient and LMF everywhere (Fig. 2b), and the sC$_T$ contribution reduces the correlation coefficient and LMF everywhere (Fig. 2c).

The pattern of the correlation coefficient and LMF depends mainly on the regional balance between the SST and sC$_T$ contributions. The direct contribution from SST to the correlation coefficient of SST and [H$^+$] is everywhere positive because an increase in temperature directly causes a rise in [H$^+$][28] (Fig. 2b). Thus, positive anomalies in SST also cause positive anomalies in [H$^+$], thereby increasing the likelihood and LMF of compound MHW-OAX events. Conversely, the contribution of sC$_T$ to the correlation coefficient is everywhere negative (Fig. 2c) because SST and sC$_T$ anomalies are everywhere negatively correlated (Supplementary Fig. 3f)[40]. Negative anomalies of sC$_T$ during MHWs (high SST) thus reduce [H$^+$] and reduce the likelihood and LMF of compound MHW-OAX events. The regionally varying magnitude of the positive SST and negative sC$_T$ contributions (Fig. 2b, c) is mainly determined by the ratios between the variabilities in SST and sC$_T$

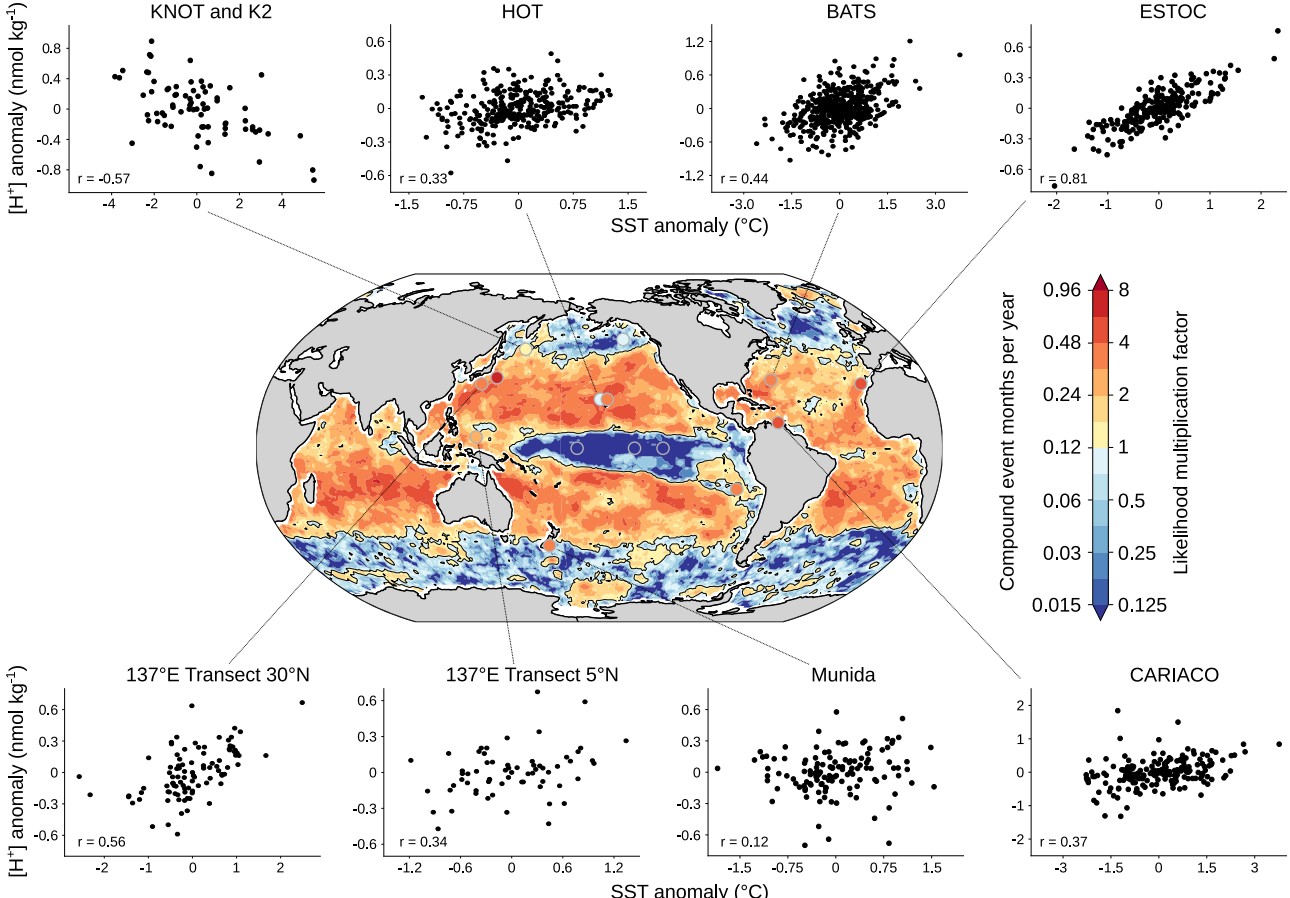

**Fig. 1 | The observation-based likelihood multiplication factor of compound MHW-OAX events over the years 1982–2019.** Map of the likelihood multiplication factor (LMF) based on the global monthly gridded observation-based SST and surface [H⁺] data. The baseline period 1982–2019 was used to define the extreme events. Warm colors indicate LMF > 1 and cold colors indicate LMF < 1. These regions are separated by the thin gray contour line. The color bar also indicates the respective number of compound event months per year. The colored dots indicate the location and the estimated LMFs from 15 observation stations (see "Methods"). SST and [H⁺] anomalies and the Pearson correlation coefficient (*r*) from eight SST and [H⁺] time series are shown around the map. All data were linearly detrended prior to the analysis.

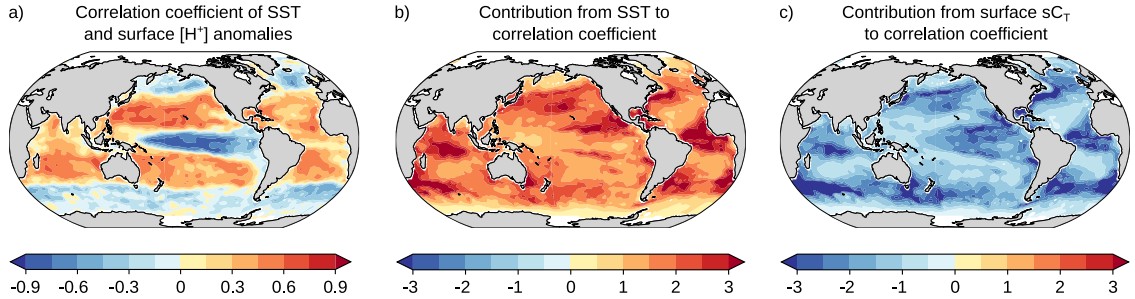

**Fig. 2 | The observation-based correlation coefficient of sea surface temperature and [H⁺] anomalies and its drivers from 1982 to 2019. a** Pearson correlation coefficient of sea surface temperature (SST) and surface [H⁺] anomalies. The contributions from **b** variations in SST and **c** variations in salinity-normalized $C_T$ to the correlation coefficient (the sum of panels **b** and **c** approximately equals to panel **a**; see "Methods"). The data were linearly detrended prior to the analysis.

anomalies and the variability in [H⁺] anomalies in the respective region (Eq. (5) and Supplementary Fig. 3).

To understand the LMF pattern, it is thus essential to understand the processes that cause variability in SST and $sC_T$. Here we qualitatively describe the contributions of the different processes, but a more quantitative understanding is needed in subsequent studies. Variability in SST and $sC_T$ results from changes in circulation, mixing, and air–sea fluxes[41,42]. Variability in surface $sC_T$ is also caused by changes in biological activity, most importantly net primary production[42].

Positive anomalies in SST, such as during MHWs, are often connected to negative anomalies in $sC_T$ as the aforementioned physical and biogeochemical processes often lead to opposite changes in temperature and $sC_T$. For example, weaker surface winds and a deepening thermocline in the central and eastern tropical Pacific during El Niño conditions drive high sea surface temperatures[5,7,43,44] but at the same time low $sC_T$ due to reductions in mixing and upwelling of colder $sC_T$-rich waters. Likewise, poleward advection of warm, low $C_T$ waters in the western boundary currents[45] lead to opposite changes in

a)     Correlation coefficient of SST
       and chlorophyll anomalies

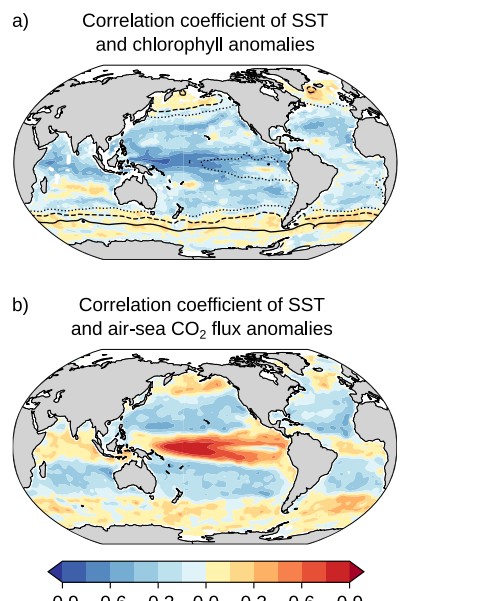

b)     Correlation coefficient of SST
       and air-sea CO$_2$ flux anomalies

-0.9  -0.6  -0.3   0.0   0.3   0.6   0.9

**Fig. 3 | Correlation of sea-surface temperature anomalies with chlorophyll and air-sea CO$_2$ flux anomalies.** The observation-based correlation coefficient of monthly sea surface temperature anomalies and monthly (**a**) chlorophyll concentration anomalies over 1998–2018 and **b** air–sea CO$_2$ flux anomalies over 1982–2019 (see "Methods"). The data were linearly detrended prior to the analysis. A negative correlation between SST and chlorophyll anomalies indicates that chlorophyll, and possibly net primary production, is reduced and hence C$_T$ enhanced during MHWs. A negative correlation between SST and air–sea CO$_2$ flux anomalies indicates that C$_T$ is reduced during MHWs. The contour lines in **a** display mean nitrate concentrations from the World Ocean Atlas 2018 (dotted: 3 µmol kg$^{-1}$, dashed: 10 µmol kg$^{-1}$, solid: 20 µmol kg$^{-1}$). Net primary production is here approximated by observation-based chlorophyll concentration[48], although chlorophyll is not always correlated with net primary production, particularly in subtropical regions[49].

temperature and sC$_T$. The sign of the change in net primary production during positive SST anomalies depends on the regional surface nutrient availability and temperature. In warm, nutrient-poor regions (40°S–10°S; 10°N–40°N), high-temperature anomalies may reduce the nutrient supply from mixing due to an increase in thermal stratification, causing a reduction in chlorophyll (Fig. 3a), possibly co-occurring with a reduction in net primary production, and a coincident increase in sC$_T$[20,43,46]. In colder high-latitude regions (>40° latitude), however, where nutrients are more abundant[47], increasing temperatures can stimulate phytoplankton growth and can reduce the light limitation through a shoaling of the mixed layer and possible increased short-wave radiation[44] and are thus associated with higher chlorophyll and primary production (Fig. 3a)[20,46] and reduced sC$_T$ (see contour lines for mean nitrate concentrations in Fig. 3a).

The resulting correlation coefficient of SST and [H$^+$] anomalies and hence the LMF (Fig. 2a) depends on the regional balance between these opposing contributions from variations in SST and in sC$_T$. In the tropical Pacific, for example, the correlation of SST and [H$^+$] anomalies is negative (Fig. 2a) corresponding to a low likelihood of MHW-OAX events (LMF < 1) (Fig. 1), although the direct effect of temperature (Fig. 2b) and the indirect effect of suppressed chlorophyll and biological production (Fig. 3a) increase the likelihood of compound MHW-OAX events in this region. Thus, the reduction in sC$_T$ and hence [H$^+$] during MHWs in this region due to suppressed upwelling that coincides with a deepening thermocline must be large enough to overcompensate the positive temperature and biology contributions to [H$^+$] and ultimately result in an LMF below 1. In the subtropical gyres, however, the correlation of SST and [H$^+$] anomalies is positive,

resulting in a high likelihood of MHW-OAX events (LMF > 1) (Fig. 2a). There, the combined positive effect from enhanced temperature (Fig. 2b) and suppressed chlorophyll (Fig. 3a) on [H$^+$] must therefore be larger than the negative effect from circulation and mixing. In the high latitudes, the correlation coefficient becomes negative again and compound MHW-OAX events become less likely (Fig. 2a). In these colder waters, increases in chlorophyll (Fig. 3a) and changes in circulation and mixing reduce sC$_T$ and hence [H$^+$] during MHWs more than temperature increases it and cause a lower LMF (Fig. 2b).

In addition to circulation, mixing, biological activity, and the direct temperature effect on [H$^+$] that determine the sign of the correlation coefficient of SST and [H$^+$] anomalies, the magnitude of the correlation and the LMF is further modulated by the changes in air–sea CO$_2$ flux (Fig. 3b). Globally, [H$^+$] and $p$CO$_2$ anomalies are strongly positively correlated ($r = 0.96$). In regions where [H$^+$] is usually increased during high temperatures, $p$CO$_2$ is also increased, resulting in an outgassing of CO$_2$ and a reduction in sC$_T$ and [H$^+$]. In contrast, air–sea CO$_2$ flux increases [H$^+$] in regions where [H$^+$] is usually decreased during high temperatures (Fig. 3b). Consequently, air–sea CO$_2$ flux reduces correlations between SST and [H$^+$] where they are positive, and it increases them where they are negative.

### Changes in MHW-OAX event occurrence with climate change
The occurrence of MHW-OAX days is projected to change with climate change (Fig. 4) and the capacity for marine organisms to adapt to the extreme events will vary. Changes in compound MHW-OAX occurrence arise from an increase in the mean state of temperature and [H$^+$], from changes in the variability of temperature and [H$^+$], as well as changes in the bivariate tail dependence of temperature and [H$^+$]. To consider different adaption capabilities of organisms and ecosystems, we define changes in the number of MHW-OAX days per year with respect to three different baselines[7,9,21]: relative to a fixed preindustrial baseline, relative to a shifting-mean baseline, and relative to a fully adapting baseline (see "Methods"). When defined with respect to a fixed preindustrial baseline, the largest changes in the number of MHW-OAX days by far are expected from long-term ocean warming and acidification trends[3,9,50]. This baseline is chosen to show the overall changes in MHW-OAX occurrence and because the fixed baseline is expected to be the most relevant baseline definition in many cases, in particular when projecting impacts for less resilient and less mobile organisms such as warm water corals[11,51] or other sessile organisms that cannot adapt to long-term ocean warming and acidification or cannot relocate to favorable ocean habitats[7]. Under the shifting-mean baseline, long-term warming and acidification trends are removed. Hence, extremes are defined as extreme deviations from the mean conditions that themselves change over time[7,9,21,52–54]. Changes in compound MHW-OAX event occurrence are here mainly caused by changes in temperature and especially [H$^+$] variability[7,9]. This baseline is chosen to analyze the role of changes in variability and to quantify the stress for organisms due to increases in extreme deviations from the mean conditions. It is most meaningful for organisms that may adapt to the long-term warming and acidification trends[55]. Furthermore, it may also be more meaningful for mobile species, such as fishes or marine mammals, because these species may relocate along gradients in the mean conditions but may still be impacted by more frequent variability-driven extremes[7,56,57], in particular if relocation is not possible on the short timescales of individual events. In addition to the two above-mentioned definitions, we also assess changes in the number of MHW-OAX days relative to a fully adapting baseline[58]. Under this baseline definition, the univariate extreme-event likelihood (e.g., the likelihood of individual marine heatwaves or ocean acidity extreme events) does not change. Instead, changes in compound MHW-OAX likelihood only arise from changes in the tail dependence between temperature and [H$^+$], altering the likelihood that MHWs and OAX events occur together. This definition is chosen to gain additional

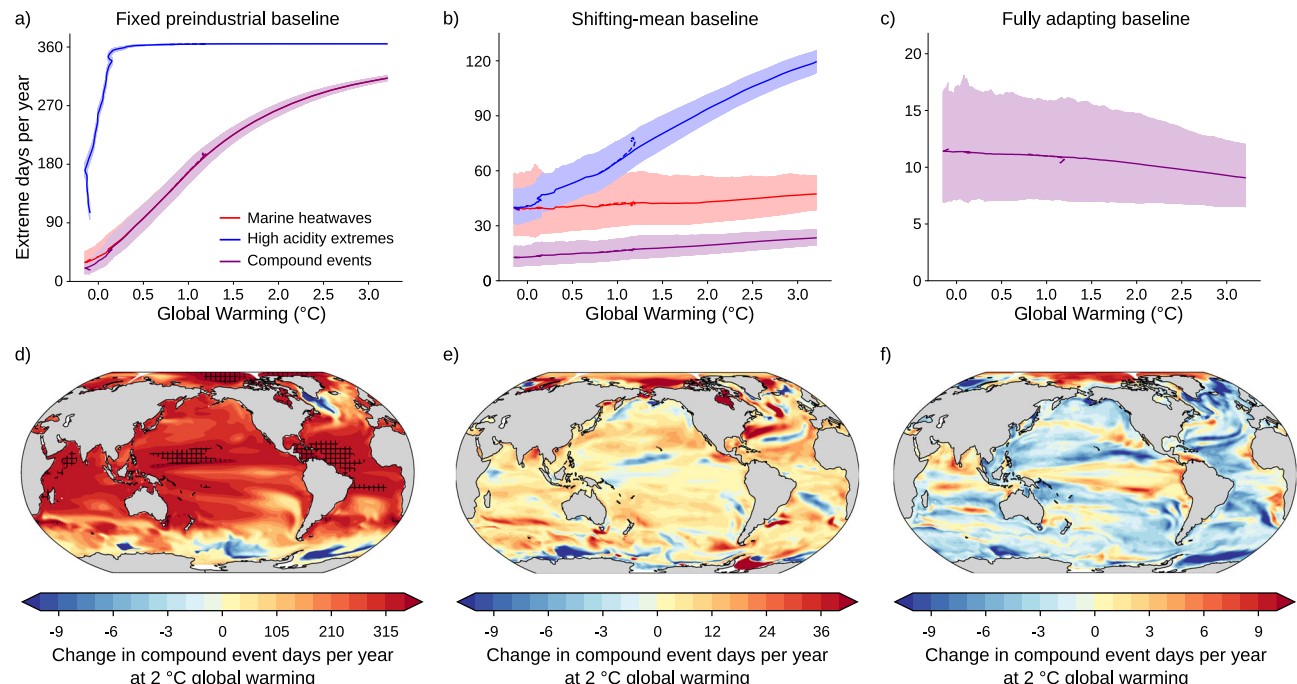

**Fig. 4 | Projected changes in the number of MHW-OAX event days per year under global warming. a–c** Global-mean number of yearly extreme-event days relative to global warming levels with respect to preindustrial conditions for MHWs (red lines), OAX events (blue lines), and compound MHW-OAX events (purple lines) for **a** fixed preindustrial baselines, **b** shifting-mean baselines, and **c** fully adapting baselines. The time series are smoothed with a 21-year running mean filter. Thick lines display the ensemble means and shaded areas depict the 10th and 90th percentile ranges of the 30 ensemble member simulations over the 1861–2100 period following the RCP8.5 scenario during the period 2006–2100. The dashed lines in **a–c** show ensemble-mean changes relative to warming levels under the RCP2.6 scenario. Differences between the RCP8.5 and RCP2.6 greenhouse gas scenarios are barely visible because the scenario spread is much smaller than the ensemble spread, indicating that the projected changes in MHW[3], OAX, and compound MHW-OAX events under global warming are independent of the warming path. **d–f** Regional changes in compound event days relative to the preindustrial period at 2 °C global warming for **d** fixed preindustrial baselines, **e** shifting-mean baselines, and **f** fully adapting baselines. Hatching in **d** indicates areas with year-round compound events (i.e., more than 360 days per year). Only results of the high-emissions scenario RCP8.5 scenario are shown in **d–f**, because the changes in globally averaged MHW-OAX occurrence are independent of the warming path.

insights about the drivers of changes, i.e., to identify the impact of changes in statistical dependence between SST and [H⁺] anomalies on MHW-OAX event likelihood.

Relative to a fixed preindustrial baseline (see "Methods"), [H⁺] reaches a near-permanent extreme-event state of more than 360 days per year already at global warming of 0.3 °C when atmospheric $CO_2$ exceeds 340 ppm (Fig. 4a), due to the large increase in mean [H⁺] compared to the natural variability in [H⁺] anomalies. A near-permanent [H⁺] extreme state causes, by definition, all MHW days to be also MHW-OAX event days (Fig. 4a). As a result, the increase in compound MHW-OAX event days per year is mainly determined by the increase in MHWs and therefore long-term ocean warming[3,50]. The occurrence of MHW-OAX event days per year is simulated to have increased 14-fold at 1 °C global warming, from 12 days per year on average at preindustrial to 167 days per year (165–169, 90% confidence interval) (Fig. 4a). Under continued global warming, MHW-OAX occurrence is projected to increase to 265 (263–266; 22-fold increase) days per year for 2 °C warming and to 307 (307–308; 26-fold increase) days per year for 3 °C. The largest increases in MHW-OAX event days per year are projected in the tropical regions of the Atlantic, the western Pacific, and the Indian Ocean (Fig. 4d). There, increasing temperatures exceed the relatively small natural variability earlier than in most other places[59] and thus lead to relatively larger increases in MHW-OAX event days per year and even near-permanent MHW-OAX events (hatched area in Fig. 4d). These near-permanent MHW-OAX events are projected to occur in 42 (42–43)% of the ocean surface area under 3 °C warming but can be largely avoided under 2 °C warming (10 (9–10)%). In regions where SST is projected to decrease over the

21st century, such as in the North Atlantic south of Greenland[60,61] and parts of the Southern Ocean[62,63], the number of MHW and compound MHW-OAX event days per year decreases (Fig. 4d). The decrease in MHW-OAX days occurs as a result of the strong decrease in MHW days and despite the co-occurring transition to near-permanent OAX events.

When defining extreme events relative to a shifting-mean baseline (see "Methods"), the occurrence of compound MHW-OAX event days is also projected to increase (Fig. 4b, e), from 12 compound event days per year at preindustrial to 19 (19–20) days per year at 2 °C warming (1.6-fold increase), and to 23 (22–23) days per year under 3 °C warming (1.9-fold increase; Fig. 4b). As opposed to the fixed baseline, the increase in MHW-OAX extreme-event occurrence under the shifting-mean baseline definition is mainly caused by a rising number of OAX events days (Fig. 4b) due to enhanced [H⁺] variability in waters with more $C_T$[9,64]. The strongest increase in compound MHW-OAX event days per year is projected for the Arctic Ocean north of 66 °N (Fig. 4e), where the reduction of sea ice leads to large increases in temperature[3] and [H⁺][9,65] variability. Although the increase in compound MHW-OAX events days per year under the shifting-mean baseline is highest there, the increase at 2 °C warming is still on average only 7% of the increase under the fixed baseline. In most other regions, the ratio between increases under the shifting-mean baseline and the fixed baseline is even smaller. An exception is the western boundary current region of the North Atlantic, where relatively large shifting-mean baseline changes exceed 10% of the fixed-baseline changes. The overall much lower increase in MHW-OAX days under a shifting-mean baseline (caused by changes in variability) than under a fixed baseline (caused by changes in mean and in variability) reflects the dominant role of

mean changes for the evolution of MHW-OAX events under a fixed baseline.

When defining extreme events relative to a fully adapting baseline (see "Methods"), the occurrence of MHW-OAX event days is projected to slightly decrease, from 12 days per year at preindustrial to 10 (10–11) days per year under 2 °C global warming and to 9 (9–10) days per year under 3 °C global warming (Fig. 4c). The decrease in MHW-OAX event days per year relative to the fully adapting baseline is equivalent to a reduction in the numerator of the LMF (see Eq. (1)) between the two periods and indicates that SST and [H$^+$] anomalies become less correlated. The reduction in correlation may be attributed to the relatively larger increase in the [H$^+$] sensitivity with respect to $C_T$ than in the [H$^+$] sensitivity with respect to temperature in warmer, high $C_T$ waters[66]. This relative increase in [H$^+$] sensitivity to $C_T$ is globally 75% larger than the relative increase in [H$^+$] sensitivity to the temperature at 2 °C global warming in the GFDL ESM2M simulation (Supplementary Fig. 5). Exceptions to the general decrease in correlation and MHW-OAX event days per year are upwelling regions, such as the eastern equatorial Pacific and the eastern boundary upwelling systems, where a reduction in s$C_T$ variability is simulated, and the Arctic Ocean, where an increase in SST-s$C_T$ correlation is simulated (Fig. 4f). In some regions, the changes in MHW-OAX events days per year relative to a fully adapting baseline can be of similar magnitude as those relative to a shifting-mean baseline. For example, in the subtropical North Atlantic (15°N–30°N, 65°W–25°W), the occurrence of MHW-OAX increases by 10 days per year relative to a shifting-mean baseline, while it decreases by 5 days relative to a fully adapting baseline under 2 °C global warming. This indicates that the reductions in the dependence of SST and [H$^+$] that reduce MHW-OAX event occurrence are over-compensated by an increase in [H$^+$] and SST variability, resulting in a net increase in MHW-OAX event days per year relative to a shifting-mean baseline.

## Discussion

The robustness of the results presented here depends on the quality of the underlying global gridded observation-based [H$^+$] product and the fidelity of the GFDL ESM2M model. The global observation-based product was evaluated with in situ time-series data from 15 ocean stations (panel and dots in Fig. 1 and Supplementary Tables 1 and 2). The differences between the LMFs estimated from the gridded observation-based product and the time series are insignificant at 14 of 15 stations (5% significance level) when accounting for the large statistical uncertainties (see "Methods"). Furthermore, the time-series estimates show a similar spatial pattern, corroborating our confidence in the gridded observation-based product[9]. Locally, the observation-based estimates are uncertain in the Southern Ocean, especially during austral winter, due to a lack of observational $p$CO$_2$ data[32,67]. Another potential caveat is that we derive [H$^+$], among other variables, from SST, which might cause an automatic correlation between both variables and hence gives an inaccurate representation of the LMF. However, multiple lines of evidence suggest that this is not the case. First, the LMF pattern (Fig. 1) and the correlation coefficient of SST and surface [H$^+$] anomalies based on observations (Fig. 2a) look very similar to the correlation coefficient based on simulated [H$^+$] and SST (Supplementary Fig. 6b). Second, the simulated pattern does not change either if $A_T$ is derived from simulated SST and salinity, or if directly simulated $A_T$ is used (Supplementary Fig. 8). Third and most important, the LMFs at ocean stations that provide directly measured [H$^+$] are similar to the LMFs calculated from measured SST, salinity, and $p$CO$_2$ at these stations. An additional caveat might be that our analysis of gridded observational data is limited to monthly-mean resolution, because most data are not available at higher temporal resolution. However, a comparison to the high temporal resolution model and buoy data suggests that the pattern of compound event occurrence is relatively insensitive to the temporal resolution (Supplementary

Table 1). We conclude that the used global gridded observation-based [H$^+$] product is well suited to analyze compound MHW-OAX events.

The simulations of the GFDL ESM2M model can be considered robust for two reasons. First, the simulated correlation of SST and [H$^+$] anomalies for the present-day shows good agreement with the spatial pattern of the gridded observational product, albeit with a general positive bias in the simulated correlation coefficient in the GFDL model, which is also present in other ESMs from the sixth phase of the Coupled Model Intercomparison Project (CMIP6; Supplementary Fig. 6). This bias suggests that the ESMs generally overestimate the effect of temperature on [H$^+$] or underestimate the effect of s$C_T$ variations. Second, the future projections based on fixed and shifting-mean baselines rely mostly on well-understood long-term ocean warming and acidification trends[65,68], as well as changes in [H$^+$] variability with increasing CO$_2$[9,64], although the exact numbers of MHW-OAX changes may depend on the model used. Under a fixed baseline, the projected changes in compound MHW-OAX events are mainly driven by secular trends in ocean warming and acidification, which are well simulated by the GFDL ESM2M model over the historical period[68]. The fixed-baseline projections are insensitive to the simulated positive bias in the correlation coefficient of SST and [H$^+$] anomalies after the onset of near-permanent [H$^+$] extreme events at around 0.3 °C global warming. Under a shifting-mean baseline, projected changes in MHW-OAX events stem mainly from an increase in [H$^+$] variability, which is also considered to be qualitatively robust since it is rooted in the nonlinear response of carbonate chemistry to increasing $C_T$[9,64]. However, a slight positive bias in the number of compound event days may be simulated due to the positive bias in correlation of SST and [H$^+$]. Furthermore, the projected global decline in MHW-OAX events with respect to a fully adapting baseline is likely robust, because it is simulated by all CMIP6 models (Supplementary Fig. 7), and because of the over-proportional increase in $C_T$ sensitivity of [H$^+$] compared to the temperature sensitivity under increasing CO$_2$ that is also expected from carbonate chemistry[28] (Supplementary Fig. 5). However, there is much less agreement between models on the regional scale (Supplementary Fig. 7) as these regional trends can be caused by variability changes in SST or s$C_T$, by correlation changes between SST and s$C_T$, or by a combination of these factors. Analysis of these changes in each model is beyond the scope of the paper calling for further analysis.

Here, we have analyzed compound marine heatwave and high [H$^+$] extreme events, but ocean acidification can also affect marine organisms via increases in $p$CO$_2$ or reductions in calcium carbonate saturation states. Due to the high correlation between [H$^+$] and $p$CO$_2$ anomalies ($r = 0.96$ on global average), compound events in [H$^+$] and temperature are often also compound events in $p$CO$_2$ and temperature as indicated by a very similar LMF pattern for high SST and high $p$CO$_2$ compound events (Supplementary Fig. 10). On global average, 78% of the months with SST-[H$^+$] compound events in the gridded observation-based data are also identified as months with extremely high $p$CO$_2$, potentially causing three-fold stress on ecosystems. Much larger discrepancies are found when analyzing compound low aragonite saturation state and high-temperature events. Due to the generally positive correlation of saturation state and temperature, the occurrence of such compound events is very rare (LMF = 0.03 on global average). This contrasts with the MHW-OAX events analyzed in this study, where high-[H$^+$] and high-temperature events overlap relatively often due to the positive temperature dependency of [H$^+$][69]. Thus, on global average only 2% of months with high SST-[H$^+$] compound events are also months with extremely low aragonite saturation state. That MHW-OAX events are usually not accompanied by an extremely low calcium carbonate saturation state may prevent calcifying organisms from additional stress due to impacts on calcification and shell dissolution[25]. However, the long-term declining trend in aragonite saturation state that is expected for the 21st century will likely result in more frequent compound extremes in temperature and aragonite

saturation state relative to fixed baselines and to more frequent occurrence of co-occurring high temperatures and aragonite under-saturation (where aragonite saturation state is below one) in polar and coastal oceans.

The combination of observations and models allowed the identification of hotspots of MHW-OAX events, estimate their frequency, better understand their drivers, and to project their development in a changing climate. Our results indicate that MHWs and OAX events are not independent and often occur together. This suggests that some of the observed MHWs[6] were also compound MHW-OAX events, in particular in the low-to-mid latitudes, where we find that one out of four MHWs are also compound MHW-OAX events when extremes are defined with respect to the 1982–2019 reference period. The reported impacts of some low-latitude MHWs on marine organisms and ecosystems[10] may therefore be also connected to additional stress from high acidity events[70]. Furthermore, also other co-occurring biogeochemical extreme events such as low oxygen events[21] or low net primary production events[20] may add to the stress during MHWs. To shed light on different aspects of changes in compound MHW-OAX occurrence under climate change, changes were assessed with respect to three different baselines. Compound MHW-OAX days are projected to become more frequent when considering the trends in SST and [H$^+$] as well as increases in their variabilities. When defined with respect to a fixed baseline, the occurrence of MHW-OAX events is projected to strongly increase, with unknown, potentially devastating effects on marine biota. Even if organisms can acclimate and adapt to long-term ocean warming and acidification or can relocate to favorable habitats, they may still be impacted by a 60% increase in compound MHW-OAX days under 2 °C global warming that emerges mainly from increasing variability in [H$^+$]. However, we also demonstrate that a decrease in the dependence of temperature and [H$^+$] anomalies may slightly dampen the increase in the co-occurrence of hot temperature and high acidity extremes, but this effect is small at global scale. The biological impacts of these changes in MHW-OAX events across different species and ecosystems are currently largely unknown[21]. The potential threat from rising numbers of MHW-OAX days highlights the urgent need to better understand the organism and ecosystem responses to such ocean compound events. In particular, the knowledge on the biological impacts of extreme conditions in [H$^+$] is still limited. A way forward would be to identify biologically informed thresholds for SST and [H$^+$] specific to key species of a certain region that directly relate such events to ecosystem impacts. Future studies on extreme events should also carefully choose the baseline depending on the impact which they analyze and potentially use absolute thresholds for specific species. Choosing the wrong baseline, shifting-mean for unmovable corals that are unable to adapt to the long-term trends, or fixed for fish that can migrate along gradients in mean conditions, may lead to an over-estimation or underestimation of the impact of changes in extreme events. Finally, this study also highlights the need for carbonate system observations on a high temporal and spatial resolution to assess and quantify biogeochemical compound events, particularly in coastal and under-sampled high-latitude regions.

## Methods

### Definition of extreme and compound events
Marine heatwaves (i.e., hot temperature extremes) and high acidity events (i.e., high [H$^+$] extremes) were defined to exceed the local and seasonally varying 90th percentiles for SST and surface [H$^+$], respectively. The seasonally varying 90th percentiles were calculated for each calendar month (gridded observational data) or calendar day (GFDL ESM2M model data) separately. Under this definition, extreme events have the same occurrence probability throughout the year. For the time-series data, there are often too few measurements for a given calendar month over the measurement period (sometimes only 10 or less) to calculate statistically meaningful monthly percentile

thresholds (as done for the gridded observation-based data). Therefore, one annual percentile threshold for monthly anomalies was calculated from all monthly anomalies instead of 12 thresholds for each month individually. The difference between calculating the LMF using seasonally varying thresholds and using one annual percentile threshold for the monthly anomalies is generally small. For example, the global average difference between these methods is 0.14 LMF units when using the gridded data-based product. Monthly anomalies were calculated by subtracting from each measurement value the mean of all measurements that were obtained in the same calendar month. The 90th percentile was chosen to have a sufficiently large sample size of observational data for statistical assessments. Sensitivity tests with the GFDL ESM2M model ensemble show that results are qualitatively insensitive to the choice of the percentile (e.g., 90th vs. 95th percentile; not shown). The usage of a percentile threshold allows the quantification of MHWs, OAX events, and MHW-OAXs events across locations that differ in variability. Absolute thresholds are often determined from the perspective of local impact, but a globally fixed absolute threshold is only meaningful in some regions, but not in others.

Compound MHW-OAX events are defined as the days or months when both SST and surface [H$^+$] are above the 90th percentile at the same time and location. We do not impose a criterion on the minimum duration of compound MHW-OAX events, as it is often applied for MHWs[1]. While such a criterion would overall reduce the number of MHW-OAX event days, it would not considerably change the LMF as can be seen from the insensitivity of the LMF to the temporal resolution of the data (Supplementary Fig. 9). The data for the present-day period (gridded observation-based product over the baseline period 1982–2019 and time-series data with varying observation periods (Supplementary Table 2)) were linearly detrended prior to identifying the extreme events.

Temporal changes in MHWs, OAX events, and MHW-OAX events within the large-ensemble model simulation were defined with respect to (i) fixed preindustrial baselines (called 'fixed preindustrial baseline' in Fig. 4), (ii) shifting-mean baselines ('shifting-mean baseline'), and (iii) fully adapting baselines ('fully adapting baseline'). Under a fixed preindustrial baseline, the extreme events were defined with respect to preindustrial seasonally varying 90th percentiles that were determined from a 500-y-long preindustrial control simulation. Under a shifting-mean baseline, these percentile thresholds were shifted according to the ensemble-mean changes with respect to the preindustrial mean state[9]. The ensemble mean was smoothed with a 365-day running mean filter to remove its seasonal cycle[9]. Under a fully adapting baseline, individual thresholds for SST and [H$^+$] were calculated for each day of the historical, RCP8.5, and RCP2.6 ensemble simulations. These were determined as the 90th percentiles of the 30-value ensemble distributions for that day as simulated by the respective 30-member ensemble simulation. As a result, the probability for univariate SST and [H$^+$] extreme events is constant over time. Changes in compound events can thus only arise from changes in the dependence between SST and [H$^+$].

### Analysis methods

**Extraction of global warming levels.** We quantify changes in compound events for different levels of global warming (e.g., in Fig. 4d–f for 2 °C global warming). To do so, 20-year periods were identified over which the ensemble-mean change in globally averaged atmospheric near-surface (2 m) temperature with respect to preindustrial conditions is closest to a specific global warming level. In the GFDL ESM2M model and under the RCP8.5 scenario, these periods are 2007–2026 (1 °C), 2045–2064 (2 °C), and 2075–2094 (3 °C).

**Confidence intervals and statistical tests for the LMF estimates.** Confidence intervals for the LMF estimates for the time series, gridded

data, and GFDL ESM2M ensemble data at the time-series locations (Supplementary Table 1) are derived by identifying the counted number of MHW-OAX days/months with the outcome of a binomial experiment[71], where the binomial "success" probability is given by the conditional probability of observing a MHW-OAX event given a MHW or OAX event. Here, the probability is assumed to be constant over time. Using the *binom* package for R, we calculate the Clopper–Pearson confidence interval[72] for the estimated conditional probability, which is directly proportional to the LMF since the probability for MHW and OAX events is a constant, here 0.1. The LMF confidence interval is then obtained by dividing the lower and upper bounds of the Clopper–Pearson confidence interval by 0.1. *P* values for the difference between the conditional probabilities estimated for the time series and for the gridded data as well as the GFDL ESM2M model are calculated using the two-sided Fisher's exact test[73] (*fisher.test* function for R). Fisher's exact test was chosen due to the often low count of MHW-OAX events in the time-series data[74].

**Estimation of the LMF from correlation coefficients.** The LMF of MHW-OAX events can be approximated based on the Pearson correlation coefficient of SST and $[H^+]$ anomalies. When assuming normally distributed monthly anomalies of SST and $[H^+]$, the estimated LMF ($\widehat{LMF}$) is given by

$$\widehat{LMF} = \frac{1}{0.1^2} \cdot \int_{x_{0.90}}^{\infty} dx_1 \int_{x_{0.90}}^{\infty} dx_2 \, f(x_1, x_2; r), \qquad (2)$$

with $r$ denoting the correlation coefficient of SST and $[H^+]$ anomalies. $f$ denotes the bivariate probability density function of two standard normal distributed variables

$$f(x_1, x_2; r) = \frac{1}{2\pi\sqrt{1-r^2}} \exp\left(-\frac{1}{2(1-r^2)}\left[x_1^2 - 2rx_1x_2 + x_2^2\right]\right) \qquad (3)$$

and $x_{0.90}$ is the 90th percentile of the standard normal distribution. The integration variables $x_1$ and $x_2$ represent SST and $[H^+]$ anomalies, here assumed to be normally distributed. The overall small difference between the counted LMF and the $\widehat{LMF}$ estimated from the correlation coefficient (global-mean deviation of 0.11) for the observation-based gridded data over the period 1982–2019 (Supplementary Fig. 1) suggests that the observed bivariate probability for exceeding the 90th percentile is similar to that assuming normally distributed temperature and $[H^+]$ anomalies and that the correlation coefficient is a good predictor for MHW-OAX compound events.

**Decomposition of the correlation coefficient into its drivers.** The observation-based correlation coefficient of SST and $[H^+]$ anomalies was decomposed into the contributions from the direct temperature dependence of $[H^+]$ and contributions from salinity-normalized dissolved inorganic carbon ($sC_T$) and total alkalinity ($sA_T$), as well as the remaining contribution from freshwater variations. For the salinity normalization, $C_T$ and $A_T$ were divided by the ratio of salinity (S, in practical salinity units) to temporal mean salinity. Thus, variations in $C_T$ and $A_T$ stemming from variations in freshwater (e.g., precipitation and evaporation) have no effect on $sC_T$ and $sA_T$. As the covariance is linear in its two arguments, covariance of temperature and $[H^+]$ anomalies (covar (SST, $[H^+]$)) can be expanded by replacing $[H^+]$ anomaly (denoted by $[H^+]$ for simplicity here) with a first-order Taylor expansion in the anomalies for SST, $C_T$, $A_T$, and S:

$$[H^+] \approx \frac{\partial[H^+]}{\partial SST}\Big|_{\overline{SST}, \dots} \cdot SST + \frac{\partial[H^+]}{\partial C_T}\Big|_{\overline{SST}, \dots} \cdot C_T + \frac{\partial[H^+]}{\partial A_T}\Big|_{\overline{SST}, \dots} \cdot A_T$$
$$+ \frac{\partial[H^+]}{\partial S}\Big|_{\overline{SST}, \dots} \cdot S. \qquad (4)$$

The partial derivatives $\partial/\partial x$ are evaluated at $\overline{SST}$, $\overline{C_T}$, $\overline{A_T}$, and $\overline{S}$, the temporal mean values of the drivers, using *mocsy 2.0*[75]. Here, it was assumed that the variations of the partial derivatives over the calendar months can be ignored. Salinity normalization for $C_T$ and $A_T$ is introduced. $C_T$ anomaly is replaced by $sC_T + \frac{\overline{sC_T}}{\overline{S}} \cdot S$, with $sC_T$ and S denoting the anomalies in salinity-normalized $C_T$ and salinity, and with $\overline{sC_T}$ and $\overline{S}$ denoting their temporal mean values, here again assuming that monthly-mean values for $sC_T$ and S can be replaced by the temporal mean values. The anomaly in $A_T$ is replaced analogously. Equation (4) is inserted in covar (SST, $[H^+]$). By using bilinearity of covariance and that covar $(x \, y) = \sigma_x \sigma_y \cdot r_{x,y}$, with $\sigma_x$ denoting the standard deviation and $r_{x,y}$ denoting the Pearson correlation coefficient, one obtains

$$r_{SST,[H^+]} = \frac{\partial[H^+]}{\partial SST}\frac{\sigma_{SST}}{\sigma_{[H^+]}} + \frac{\partial[H^+]}{\partial C_T}\frac{\sigma_{sC_T}}{\sigma_{[H^+]}}r_{SST,sC_T} + \frac{\partial[H^+]}{\partial A_T}\frac{\sigma_{sA_T}}{\sigma_{[H^+]}}r_{SST,sA_T}$$
$$+ \left(\frac{\overline{sC_T}}{\overline{S}}\frac{\partial[H^+]}{\partial C_T} + \frac{\overline{sA_T}}{\overline{S}}\frac{\partial[H^+]}{\partial A_T} + \frac{\partial[H^+]}{\partial S}\right)\frac{\sigma_S}{\sigma_{[H^+]}}r_{SST,S} + \text{residual}.$$
$$(5)$$

The terms on the right-hand side of Eq. (5) represent the contributions from SST, $sC_T$, $sA_T$, and freshwater variations (from left to right). The residual of the decomposition (simulated correlation coefficient minus sum of decomposition terms) for the gridded data product over the period 1982–2019 (Fig. 2 and Supplementary Fig. 2) is smaller than 0.1 correlation coefficient units for 99% of the ocean surface (Supplementary Fig. 2c). The freshwater term only has a small imprint on SST-$[H^+]$ correlation (Supplementary Fig. 2b) since the direct effect from salinity variations on $[H^+]$ is small and because its effects on $C_T$ and $A_T$ largely compensate each other. Likewise, the contribution from $sA_T$ is comparably small (Supplementary Fig. 2a). The approximation of the correlation coefficient by the contributions of SST and $sC_T$ is precise to 0.1 units in 60% of the ocean surface and to 0.4 in 92% of the ocean surface. The sign of the approximated correlation coefficient is correct in 93% of the ocean surface.

**Observation-based data**
**Global sea surface temperature and [H+] data.** Global monthly SST data on a grid with 1° horizontal spacing from 1982 to 2019 from the version 4.2.1 (EN4.2.1) dataset developed by the Met Office Hadley Centre[33] were used. EN4 is a gridded product that is based on temperature and salinity profiles, with World Ocean Database[76] being the main source of data. SST is here defined as mean temperature over the uppermost 10 m. The global gridded observation-based $[H^+]$ (total scale) dataset covering the period from 1982 to 2019 was reconstructed in two steps following the method outlined in ref. 9. In a first step, the SST and surface salinity fields from EN4.2.1 were used to derive total surface alkalinity ($A_T$) using the *LIARv2* total alkalinity regression algorithm[38]. In a second step, a monthly $[H^+]$ field at 1° horizontal grid resolution was derived using the CO2SYS carbonate chemistry package[77] from the monthly, gridded surface salinity, SST, and $A_T$ and from the interpolated global surface $pCO_2$ product *MPI-SOMFFN*[32] that covers the period 1982–2019 with monthly-mean temporal resolution and is based on data from the *Surface Ocean CO2 Atlas version 4*[78].

The accuracy of the LIARv2 algorithm was tested with the GFDL ESM2M model output by comparing directly simulated $A_T$ ($A_T^{sim}$) with the $A_T$ that was estimated by the LIARv2 algorithm from simulated SST and surface salinity for one ensemble member over the 1982–2019 period ($A_T^{LIAR}$). The comparison yields a root-mean-square error between $A_T^{sim}$ and $A_T^{LIAR}$ of 32 mol kg$^{-1}$. This difference between the $A_T^{sim}$ and $A_T^{LIAR}$ translates into differences in $[H^+]$ and hence into differences between the correlation coefficient of SST and $[H^+]$ anomalies and between the LMF. These differences between the correlation coefficients when using $A_T^{sim}$ and $A_T^{LIAR}$ are below 0.1 units in 99.5% of

the surface ocean and the difference in the LMF is below 0.5 in 96.5% of the surface ocean area (Supplementary Fig. 8), indicating that our estimated observation-based $A_T$ from SST and sea surface salinity is accurate enough to be used in the calculation of observation-based [H$^+$]. The same comparison was made for ocean stations where direct $A_T$ measurements are available. The LMF using direct $A_T$ measurements slightly differ from the LMF using estimated $A_T$ (not shown) but remain within the large uncertainty bounds that exist at these stations due to the relatively short observational period, indicating that our LMF results are likely insensitive to the exact choice of estimated $A_T$.

GFDL ESM2M model output suggests that compound MHW-OAX events are often shorter than one month (Supplementary Fig. 4). Nevertheless, the LMF of present-day compound events can be approximated based on monthly (observational-based) data. Comparison of the LMF obtained from monthly-mean model output to that from daily-mean model output for the period 1982–2019 yields a root-mean-square error of 0.2 (Supplementary Fig. 9). This deviation results mainly from a small bias in the LMF from monthly-mean data compared to daily-mean data (global-mean LMF of 2.7 in monthly-mean data vs. 2.9 in daily-mean data; Supplementary Fig. 9c). As such, using monthly data may slightly underestimate the occurrence of MHW-OAX events.

**Time series of in situ SST and [H$^+$] data.** In situ observations from the KNOT and K2 stations in the north Pacific, the Hawaii Ocean Time-series (HOT), the Bermuda Atlantic Time-series Study (BATS), the European Station for Time series in the Ocean Canary Islands (ESTOC), the 137 °E transect at 30°N and at 5°N, the Munida time series, the CARIACO time series, and from seven autonomous open-ocean surface buoys were used in this study (Fig. 1 and Supplementary Table 2). The KNOT and K2 station data were combined due to their spatial proximity (44°N, 155 °E vs 47°N, 160°E) resulting in a total of 15 time series. For the seven autonomous open-ocean surface buoys, monthly means were calculated from the 3-hourly data to match the approximate measurement frequency of the remaining stations. For completeness, the buoy data was also aggregated to daily means (Supplementary Table 1). Only buoys providing at least 5 years of data were analyzed, being KEO (Kuroshio Extension Observatory), Papa (Ocean Station Papa), Stratus, TAO125W, TAO140W, TAO170W (NDBC Tropical Atmosphere Ocean 0°N at 125°W, 140°W, and 170°W, respectively), and WHOTS (Woods Hole Oceanographic Institution Hawaii Ocean Time-series Station)[79].

[H$^+$] was calculated from measured pH at in situ temperature where available. If pH at in situ temperature was not available, [H$^+$] was calculated from sea surface temperature, salinity, $A_T$, phosphate, and silicate and, depending on the time series, from either measured $C_T$ or $p$CO$_2$ (Supplementary Table 2). If calculated pH was not provided by the dataset, *mocsy 2.0*[75] was used to calculate [H$^+$] when $C_T$ was measured and CO2SYS[77] was used when $p$CO$_2$ was measured. CO2SYS was used in the latter case because *mocsy 2.0* does not allow the calculation of [H$^+$] from pCO$_2$. However, differences between the *mocsy 2.0* and CO2SYS for [H$^+$] calculations are small[80]. Where not available, mean silicate and phosphate concentrations were taken from the closest grid cells of World Ocean Atlas 2018[81,82]. For the seven autonomous open-ocean surface buoys, $A_T$ was estimated from measured sea surface temperature and salinity using the *LIARv2* algorithm[38].

**Chlorophyll concentration, nitrate, and air–sea CO$_2$ flux data.** Daily-mean data-assimilated chlorophyll concentration output (Fig. 3a) from the NASA Ocean Biogeochemical Model (version NOBM.R2020.1)[83] over the period 1998–2018 was used in this study. The model assimilates satellite chlorophyll data from the Sea-viewing Wide Field-of-view Sensor, the Moderate Resolution Imaging Spectroradiometer Aqua and the Visible Infrared Imaging Radiometer Suite and provides daily data within the mixed layer. Monthly-mean values were calculated prior to the analysis. Mean nitrate concentration data from the World Ocean Atlas 2018[81,82] was also used in Fig. 3a. The observation-based monthly air–sea CO$_2$ flux data product over the period 1982–2019 shown in Fig. 3b is based on the interpolated $p$CO$_2$ product[32,84] that was used to derive the gridded [H$^+$] product.

**GFDL ESM2M large ensemble**
**Model setup and simulations.** The fully coupled GFDL ESM2M Earth system model[36,37], developed at the NOAA Geophysical Fluid Dynamics Laboratory, was used in this study. The physical ocean component of the model, the Modular Ocean Model (version 4p1[85]), has a nominal 1° horizontal resolution with increasing resolution near the equator, and 50 vertical levels with vertical resolution decreasing from 10 m at the surface to 300 m at 5000 m depth. This physical ocean model is coupled to the ocean biogeochemistry model TOPAZv2, which simulates 30 tracers and three phytoplankton groups. Zooplankton grazing is implicitly simulated. The ocean carbonate chemistry routines are based on the OCMIP2 protocol[86].

A 30-member ensemble simulation from 1861 to 2100 was performed. The large-ensemble size allows for a robust assessment of changes in compound events (Fig. 4a–c) and is necessary to assess changes in compound events relative to a shifting and fully adapting baseline. The 1861–2005 period was forced with prescribed historical greenhouse gas and aerosol concentrations[87]. The period from 2006 to 2100 was simulated with greenhouse gas and aerosol concentrations from the low-emissions high mitigation concentration pathway RCP2.6[88] and from the high-emissions no mitigation concentration pathway RCP8.5[89]. However, the main global-mean results shown in Fig. 4 do not depend on the warming path and are hence independent from the emission scenario. As a result, Fig. 4 mainly shows results for the RCP8.5 scenario. However, global-mean changes for the RCP2.6 scenario are also shown in Fig. 4a–c. A 500-year preindustrial control simulation was also performed to calculate the fixed preindustrial baseline. The initialization procedure of the large-ensemble simulation is described in Burger et al.[9]. Daily-mean ocean temperature and [H$^+$] (on the total scale) output from the surface layer of the ocean model (0–10 m) was analyzed.

**Model evaluation.** The capability of the GFDL ESM2M model to simulate the mean state and long-term changes in ocean biogeochemistry and in particular OA extremes, as well as MHWs, has been extensively evaluated. The model has been shown to capture the mean state in [H$^+$][68] and its mean seasonal cycle[64], as well as changes in [H$^+$] seasonality over the last few decades[9]. Furthermore, the model reproduces the positive long-term trend in global-mean SST[59] and annual number of MHW days over the satellite period[3].

In addition, the model's performance in simulating compound MHW-OAX events was evaluated here. The LMF of MHW-OAX events estimated from model output agrees with the estimates from the time-series data in 14 of 15 cases (differences are insignificant under a 5% significance level; Supplementary Table 1). As for the gridded observation-based data product, we find a significant difference for the HOT station. However, the agreement with the WHOTS time series that is measured at the same location may indicate that the difference may be due to the temporal resolution of the data: HOT consists of point measurements while the GFDL ESM2M model data and gridded data product provide daily and monthly-mean values, as for WHOTS where monthly-mean values were calculated from 3-hourly measurements. Moreover, the simulated LMF patterns for MHW-OAX events from the monthly model output are similar to the patterns estimated from the gridded observation-based, monthly data product (compare Fig. 1 and Supplementary Fig. 9b). However, the GFDL ESM2M model overestimates the occurrence of MHW-OAX events at the global scale (LMF of 2.7 in the GFDL ESM2M model versus 1.8 in the observation-based product). Locally, the LMF overestimation is especially pronounced in

the high latitudes such as the Southern Ocean (40°S–81°S; LMF of 1.9 vs 0.7), the North Pacific (40°N–66°N; 1.9 vs 1.0) and the North Atlantic (40°N–66°N; 1.8 vs 0.8). In the Southern Ocean, however, the observation-based estimates are also uncertain due to a lack of observational data, especially during austral winter for $pCO_2$[32,67]. Similarly, a bias of too large correlation coefficients of SST and [H$^+$] anomalies is evident in all here analyzed CMIP6 model simulations (Supplementary Fig. 6), suggesting that comprehensive ESMs systematically overestimate the effect of temperature on [H$^+$] (contributing positively to correlation) compared to the effect of $sC_T$ variations (contributing negatively to correlation). Projections based on fixed baselines are insensitive to this positive bias in correlation because MHWs always coincide with OAX events already at 0.3 °C global warming when OAX events become near-permanent. However, projections relative to shifting-mean baselines may be slightly positively biased.

When analyzing the correlation coefficient of SST and [H$^+$] anomalies, the GFDL ESM2M model performs well compared to most CMIP6 models (Supplementary Table 3), in particular in the subtropics and tropics (red and gray lines in Supplementary Fig. 6k). At the global scale, the GFDL ESM2M model outperforms most of the CMIP6 models (red vs. gray stars in Supplementary Fig. 6l).

Overall, the agreement between the simulated present-day compound MHW-OAX event pattern and the observation-based pattern, in addition to the model's fidelity in simulating recent trends in MHW characteristics[3] and [H$^+$] seasonality[9], gives high confidence in the use of the GFDL ESM2M model for analyzing patterns and trends in compound MHW-OAX events at the global scale.

## Data availability
The data underlying the analyses in this study have been deposited in the Zenodo repository under https://doi.org/10.5281/zenodo.6655451.

## Code availability
The code to generate the results from the underlying data has been deposited in the Zenodo repository under https://doi.org/10.5281/zenodo.6655451.

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

## Acknowledgements

This study has been supported by the Swiss National Science Foundation (No. PP00P2_198897), the European Union's Horizon 2020 research and innovation program under grant agreement No. 820989 (project COMFORT, Our common future ocean in the Earth system-quantifying coupled cycles of carbon, oxygen, and nutrients for determining and achieving safe operating spaces with respect to tipping points), and No. 862923 (project AtlantECO, Atlantic Ecosystems Assessment, Fore-casting & Sustainability). The work reflects only the authors' view; the European Commission and their executive agency are not responsible for any use that may be made of the information the work contains. The GFDL ESM2M large-ensemble simulations have been conducted at the Swiss National Supercomputing Centre (CSCS). We thank Donat Hess for initial analysis, and Fortunat Joos and Jakob Zscheischler for discussions. We further thank Daisuke Sasano and Melchor González Dávila for providing ocean time-series data. We acknowledge the World Climate Research Programme, which, through its Working Group on Coupled Modeling, coordinated and promoted CMIP6. We thank the climate modeling groups for producing and making available their model output, the Earth System Grid Federation (ESGF) for archiving the data and providing access, and the multiple funding agencies that support CMIP6 and ESGF.

## Author contributions

The initial study was designed by F.A.B. and T.L.F. The ensemble simulations and analysis were conducted by F.A.B. F.A.B. produced the figures, with help from T.L.F. and J.T. All authors contributed ideas and discussed results. F.A.B. wrote the initial draft and all authors contributed to the writing.

## Competing interests

The authors declare no competing interests.
