## [Peer Review File · Nature Communications]

Compound marine heatwaves and ocean acidity extremesEditorial Note: This manuscript has been previously reviewed at another journal that is not operating a transparent peer review scheme. This document only contains reviewer comments and rebuttal letters for versions considered at *Nature Communications*.

REVIEWER COMMENTS

Reviewer #4 (Remarks to the Author):

review of Burger et al.

The paper provides information on MHW and OAX events and their co-occurrence and will be of broad interest to researchers investigating biological impacts of extreme events and others working on characterizing the events using observations. Detailed information is provided by the authors, particularly in response to the extensive previous reviews. This is a timely publication as more focus is directed at understanding multi-stressor impacts on marine life.

The reasons for the different sections are at times buried in the text, and some sections required many reads and back tracking to try and understand the relevance. I agree with their use of ocean acidity in response to a previous reviewer's comment. This is a widely used term and the acid-base divide at $\text{pH} = 7$ is really for infinitely dilute solutions. Ocean acidity is used by many to refer to changes in the $[\text{H}^+]$ and seems appropriate for this paper.

The concept of the LMF calculation (equation 1) and how it is used in the results and discussion are good. However, I cannot follow the LMF description in lines 85-86. The LMF calculation is at the center of this paper and it may just be my lack of background in statistics. The Zscheichler and Seneviratne (2017) paper referenced for the LMF calculation helps a little, but I hope the authors will be able to make the statement on these lines clearer. Understanding how to define co-occurrence of extreme events is becoming increasingly important.

Marine biota respond differently depending on their resilience, the severity/duration of extreme events, and capacity to adapt to change. Lines 548-555 in the supplement does mention the model results indicate events are "often shorter than one month (Ext. Data, Fig 4)". This is useful information for determining observing needs and while the comparison with observation-based (monthly) and monthly average model output generally agrees (lines 552-554), it would be useful to add this assessment is for the open ocean. Coastal conditions may lead to different results. If these events are due to upwelling in coastal waters, it may be short term and decrease the LMF (e. g. analogous to the equatorial Pacific). On the other hand, ocean heat waves may substantially increase LMF by influencing $[\text{H}^+]$ in oligotrophic and stratified regions (e. g. western boundary currents). The authors should also consider including a statement in the conclusions of the importance of developing high frequency observing for under sampled open-ocean (e. g. polar oceans and Southern Hemisphere) and coastal regions to better define extreme events and the associated biological responses. Otherwise, it seems like the connection back to the observations used to determine LMF and ultimately constrain the models is missing.

Abstract: The abstract does not state the study is for the open ocean. This is mentioned in Line 66 of the Introduction and should be in the abstract. Also, mention of MHW-OAX events are 1.8 in 100 months does not indicate this is for monthly gridded products as described in lines 96 -104 of the results section. The first part of the abstract has extreme events in months per 100 months (using monthly gridded data products) and the last part of the abstract changes to events in days per year. It would be helpful to the reader to identify that daily model output was used for the latter results. I would also add to line 24 the increase to 265 days under 2C global warming is relative to a fixed pre-industrial baseline. The sentence in line 25 mentioning the 60% increase in events for a shifting mean baseline may also benefit by including something like "...a 60% increase (12 to 19 days)....". This is to reduce confusion around the shift from months per 100 months, to days per year, to percent change - all of which occur in one long paragraph.

Line 51. The paper referenced as number 21 is listed in as submitted to Nature in 2020. Is it published?

Line 86. Closed off bracket is missing at end of sentence on this line. The authors should also change "daily data" in this line to "daily model output". The daily results are modelled not observed.

Line 113. The [H+] in the eight time series sites are not all directly measured as far as I am aware. SST is directly measured and [H+] is often calculated from carbon system data. Changing from "...eight directly measured SST and [H+]...", to "...eight SST and [H+]..." would fix this statement.

Lines 152-206. The authors provide a great description on the LMF pattern change. One issue for the Arctic and Southern Ocean is there are so few data available, and the models indicate considerable regional variability (Fig 4). The potential to disrupt ecosystems is compounded by these waters being close to undersaturation, so crossing to undersaturated waters may be more significant for the ecosystems of these regions than the co-occurrence of extreme events described in the paper. here is no mention of this in the paper.

Line 209. The paragraph starting here does describe the different scenarios for extreme events, but would it not be simpler for the reader to state at the beginning what this section is about i.e. "The frequency of MHW-OAX days is projected to change with climate (Fig. 4) and the capacity for marine organisms to adapt to the extreme events will vary. Changes in MHW-OAX occurrence will arise from an increase in the mean state of temperature and [H+], from changes in the variability of temperature..."? As with other sections of this paper I find the reason for a section is often further down the page. This is a different writing style and may not be the preference of the authors, but it may make reading the paper easier.

Line 212-222. The sentences in these lines would be useful at the end of the introduction. It describes why different methods for calculating LMF were used (fixed baseline versus trend removal), but this is again another decision for the authors. For me, it makes more sense.

Line 243. In Figure 4 caption, a dashed line is mentioned, and I cannot see this apart from a small squiggle in 4b (blue dashes above the solid blue line at just over 1C warming). Is it relevant?

Line 253-, For Figure 4a a cause of the rapid rise in OAX events is mentioned in Line 259-260. I appreciate the authors are focused on OAX-MHW events. The figure 4a is the first thing that will be looked at by a reader and why the reason for the rapid increase in OAX events in not made apparent first up will confuse some readers. The following is just personal preference and if the authors do not agree, I have no issue with this. However, I find I am continually stopping reading to cross check the figures and text it takes a long time to get through the paper and this is frustrating.

"For a fixed preindustrial baseline, [H+] reaches a near-permanent extreme state of more than 360 days per year (Fig. 4a) as the seasonal variability in H+ exceeds the preindustrial values at global warming of 0.3C, or when atmospheric CO₂ exceeds 340ppm. Here, the increase in compound MHW-OAX events is mainly determined by the increase in MHWs as the near-permanent H+ extreme state causes, by definition, all the MHWs to be MHW-OAX events. For the preindustrial baseline, the occurrence of MHW-OAX events in Figure 4a is simulated to have increased 14-fold from 12 days per year on average at preindustrial to 167 days per year (165-169, 90% confidence interval) at 1°C global warming. Under continued global warming, MHW-OAX occurrence is projected to increase to 265 (263-266; 22-fold increase) days per year for 2°C warming and to 307 (307-308; 26-fold increase) days per year for 3 °C."

Lines 267-270. Is there a reference to the projected decrease in SST for waters of the Southern Ocean and near the Greenland Sea and why this would occur under global warming over the 21st century (stratification change etc)?

Line 576. Mocsy 2.0 is not the standard used to calculate the marine CO₂ system. It may be for modellers, but not observationalists. This claim does not need to be included.

Line 626 - 633. The paper discusses discrepancies between LMF estimates for the Hawaii time

series station (HOTS). Are the authors stating the LMF values for the monthly gridded values do not agree with the monthly HOTS carbon chemistry sampling, but do agree when the 3-hourly pCO₂ data is used to calculate monthly mean values? This is surprising given the HOTS site is in oligotrophic waters and the gridded data products are heavily constrained in the region by the WHOTS data and ship-based pCO₂ data. It is probably one of the best sampled sites in the ocean. Perhaps it indicates the calculation still has surprises, particularly given the limited number of time series sites.

How is the LMF data accessed?

I strongly recommend the authors add a sentence or two in the conclusions about the need to expand the observation base to better characterize change and the related biological impacts. Otherwise, the paper just ends with a "here is what we found statement", whereas the value will increase by highlighting the need to expand to coastal and under-sampled regions, particularly if the model predictions of change are correct.

Reviewer #5 (Remarks to the Author):

Compound marine heatwaves and ocean acidity extremes
Burger et al

This article has been reviewed by other reviewers, however this is my first reading of the manuscript. This paper examines the co-occurrence of marine heatwave and high [H⁺] events using a combination of gridded observations, a large ensemble climate model and selected station data. The authors show that compound events are more likely than randomly distributed compound events would be across most of the ocean (except in the tropical Pacific and high latitudes) and examine the causes for this.

This is a new and significant area of research, given the likelihood that compound stressors will produce synergistically negative effects in marine organisms. In general, this is an interesting article with some interesting and useful insights. I do however have a few concerns/questions detailed below.

General comments:

I find the reference to terms like events/frequency somewhat misleading. I suspect that most readers would use an event to refer to a continuous event with extreme conditions that can last for extended periods of time and the frequency to refer to the number such events per time period. Here an event refers to each month above the extreme threshold and frequency refers to the number of months (rather than the number of events). I feel a clearer terminology is needed

Its not clear to me that the gridded datasets based only on ship data would actually be able to capture extreme events, given that these datasets are based on sparse data. If for example you do a scatter plot of station based vs gridded SSTA or [H⁺] do you actually get a reasonable match?

I find the biological argument for a moving baseline rather unconvincing. This scenario assumes that the species is able to adapt to large changes in mean climate, but unable to adapt to much smaller changes in variability. While it is certainly interesting to separate changes in the mean from the variability to understand the physical processes at play, Im not sure what biological relevance this really has.

As you show in Fig 4a a small amount of warming means that you very quickly reach permanent H⁺ extreme conditions (presumably because H⁺ variability is so small compared to the climate change signal). This would suggest to me that the 90th percentile is not a very useful metric. Indeed if 0oC global warming represents pre industrial conditions, then we should have reached permanent H⁺ extreme by the mid to late 20th century. I think this point at least requires some discussion.

Specific comments:

98: larger than one OVER 65% of the ocean

Fig 1 for KNOT and K2 station the SSTA-[H+] correlation is negative, however the associated circle shows a positive LMF – how is this possible? The opposite seems to be the case for HOT i.e. positive correlation but -ve LMF.

120: It is hence approximated by the Pearson correlation coefficient ...

I think this terminology is confusing. The sentence makes it sound like the correlation coefficient is approximately equal to the LMF. I don't believe this is what you mean, you mean that the two are related.

126: The freshwater CYCLING term quantifies the direct impact

127: CT and AT ...

AT hasn't been defined yet

156: Positive anomalies in SST, such as during MHWs, are often connected to negative anomalies in sCT as these processes often lead to opposite changes in temperature and sCT

This sentence seems circular to me?

158: and enhanced thermal stratification

During El Nino the thermocline becomes deeper in the east, which means its harder for the deep waters to be brought to the surface the surface – I don't think this is manifest as a greater stratification.

164-165: Here is where I would refer to stratification. At mid latitudes MHWs are associated with enhanced stratification that would suppress mixing of nutrients between surface and deep layers

189: that the reduction in sCT and hence [H+] due to suppressed upwelling and mixing during MHWs ...

It will also be related to the deeper pycnocline

219: organisms that cannot adapt fast enough to ...

Your assumption here is not that they cant adapt fast enough, its that there is no adaptation (you are fixing the baseline).

225: that may ADAPT to long-term ...

225 (and also in the discussion): or can shift their distribution ...

This doesn't make sense to me. Why would a moving baseline be appropriate for mobile species?

If an organism can adapt, then it wouldn't need to move - poleward migration occurs because an organism cant adapt to warming temperature.

Fig 4, the curves appear to turn back on themselves. Is this because there is still some residual variability in the ensemble mean (i.e. temperatures don't increase monotonically)?

269: the frequency of MHWs and hence compound MHW-OAX events decreases ...

It seems strange that in regions where sst shows little change (so MHW frequency shouldn't change much) but H+ extremes become a permanent feature, the result is a decrease in compound events.

275: 1.9-fold increase

It looks like there is a 3x increase in H+ extreme days, and a small increase in MHW days, so why do we only get a 1.9x increase in compound events?

291: The reduction in correlation may be attributed to the over- proportional increase in the [H+] sensitivity with respect to CT in warmer, high CT waters

Is there a reason why this occurs?

334: anomalies FOR THE present-day

335: ... albeit with a general positive bias

376. Not sure what you mean by: allowed to localize

Methods:

I don't believe that the baseline periods have been defined for the observation (pre industrial is used for the model). MHW statistics are very sensitive to the baseline used.

406: For the time series data ...

You haven't introduced the timeseries or other datasets yet in the methods. I think it would make

more sense to start the methods with the datasets used.

412: threshold is generally small

Perhaps I have misunderstood, but I don't understand how this effect can be small. If you are using an annually fixed threshold you would only pick out summer MHW for example.

You test this on the gridded data, but I wonder if taking a global average (which includes a mix of summer and winter from different hemispheres) hides much larger single hemisphere errors.

437: The ensemble mean was smoothed with a 365-day running mean filter to remove its seasonal cycle

Doesn't this mean that your analysis doesn't separate between changes in extremes and changes in the seasonal cycle. Is this important, given that there are projected changes in SST seasonality?

440: as the 90th percentiles of the 30-value ensemble distributions for that day

How do you calculate a robust 90th percentile using only 30 values. Or do you use a window around each day of the year to increase your sample size.

488: For those that read the methods first it would be useful to define CT (sCT) and AT (sAT) again here.

Response to comments by the editor and reviewers #4 and #5

March 10, 2022

Summary:

We thank the editor and the reviewers for their overall positive evaluation of our manuscript and the many helpful comments and suggestions. In the following, we respond to their comments point-by-point and especially address the main concerns of reviewer #4 about the description of the Likelihood Multiplication Factor and the missing conclusion on future observational needs and of reviewer #5 about the suitability of the gridded dataset and the moving baseline motivation.

We hope that this revision finds your approval.

Friedrich Burger

Response to comments by Referee 4

The paper provides information on MHW and OAX events and their co-occurrence and will be of broad interest to researchers investigating biological impacts of extreme events and others working on characterizing the events using observations. Detailed information is provided by the authors, particularly in response to the extensive previous reviews. This is a timely publication as more focus is directed at understanding multi-stressor impacts on marine life.

We thank the reviewer for this positive assessment, the recognition of the novelty of the article, and the insightful comments. We provide detailed responses to all comments below.

The reasons for the different sections are at times buried in the text, and some sections required many reads and back tracking to try and understand the relevance.

We hope that our revisions have improved the readiness of the manuscript.

I agree with their use of ocean acidity in response to a previous reviewer's comment. This is a widely used term and the acid-base divide at $\text{pH} = 7$ is really for infinitely dilute solutions. Ocean acidity is used by many to refer to changes in the $[\text{H}^+]$ and seems appropriate for this paper.

We thank the reviewer for supporting the usage of the term 'ocean acidity'.

The concept of the LMF calculation (equation 1) and how it is used in the results and discussion are good. However, I cannot follow the LMF description in lines 85-86. The LMF calculation is at the center of this paper and it may just be my lack of background in statistics. The Zscheichler and Seneviratne (2017) paper referenced for the LMF calculation helps a little, but I hope the authors will be able to make the statement on these lines clearer. Understanding how to define co-occurrence of extreme events is becoming increasingly important.

We appreciate that the reviewer finds the LMF concept useful. To improve the clarity of the LMF description, we adapted lines 82-86 to:

“We here quantify the number of MHW-OAX months per year (for the observation-based data) and the number of MHW-OAX days per year (daily-mean model output), irrespective of whether these months or days belong to the same ongoing MHW-OAX event. Similarly, we define the likelihood multiplication factor (LMF) that quantifies the likelihood that a month or day is under MHW-OAX conditions relative to the expected likelihood if MHWs and OAX events would occur independently from each other. The LMF is the ratio between the observed likelihood of compound event months or days $p(\text{MHW} - \text{OAX day or month})$, calculated as the percentage of months or days that are under MHW-OAX conditions over a given period, and the theoretical likelihood of compound event months or days if SST and $[\text{H}^+]$ were statistically independent. The theoretical likelihood of compound event months or days for statistically independent variables is given by the product of the individual likelihoods of MHWs and OAX months or days $p(\text{MHW day or month}) \times p(\text{OAX day or month})$ (Zscheichler et al., 2017):

$$LMF = \frac{p(\text{MHW-OAX day or month})}{p(\text{MHW day or month}) \times p(\text{OAX day or month})} \quad (1)$$

For our definition of single extreme event months or days (90th percentile thresholds), this theoretical likelihood is $10\% \cdot 10\% = 1\%$, corresponding to 0.12 months per year for monthly data or 3.65 days per year for daily model output under MHW-OAX conditions. An LMF higher than 1 indicates that compound event

months or days occur more often than by chance and that more than 1% of all months or days are under MHW-OAX conditions. An LMF lower than one indicates a reduced likelihood of compound event months or days. As an example, if four out of 200 months were under MHW-OAX conditions, the LMF would be 2 meaning that the likelihood of a compound event month would be twice as large as under independence. However, if only one out of 200 months was under MHW-OAX conditions, the LMF would be 0.5 meaning that the likelihood of a compound event month would be only half as large as under independence.”

We now accompany the discussion of the observation-based LMF results with the absolute numbers of months per year (or days per year) under compound MHW-OAX conditions, and we also show these absolute numbers in Fig. 1 to further ease the understanding of our results.

Marine biota respond differently depending on their resilience, the severity/duration of extreme events, and capacity to adapt to change. Lines 548-555 in the supplement does mention the model results indicate events are "often shorter than one month (Ext. Data, Fig 4)". This is useful information for determining observing needs and while the comparison with observation-based (monthly) and monthly average model output generally agrees (lines 552-554), it would be useful to add this assessment is for the open ocean. Coastal conditions may lead to different results. If these events are due to upwelling in coastal waters, it may be short term and decrease the LMF (e. g. analogous to the equatorial Pacific). On the other hand, ocean heat waves may substantially increase LMF by influencing [H+] in oligotrophic and stratified regions (e. g. western boundary currents).

We agree. We now state prominently in the abstract that the assessment in this study is for the open ocean:

“Using monthly open-ocean observations, we show that...”

In addition, we extended the statement in the introduction of the main manuscript on lines 66-69 that the analysis is restricted to the open ocean:

“The analysis is restricted to the open ocean, since the high variability and locally important processes in coastal oceans, such as riverine fluxes or shelf and coastal dynamics, are neither captured by the gridded observation-based product (Landschützer et al., 2016; Carter et al., 2018) nor by the GFDL ESM2M model (Dunne et al., 2013).”

The authors should also consider including a statement in the conclusions of the importance of developing high frequency observing for under sampled open-ocean (e. g. polar oceans and Southern Hemisphere) and coastal regions to better define extreme events and the associated biological responses. Otherwise, it seems like the connection back to the observations used to determine LMF and ultimately constrain the models is missing.

As suggested, we add the following sentence to the Conclusion:

“Finally, this study also highlights the need for carbonate system observations on high temporal and spatial resolution to assess and quantify biogeochemical compound events, particularly in coastal and under-sampled high-latitude regions.”

Abstract: The abstract does not state the study is for the open ocean. This is mentioned in Line 66 of the Introduction and should be in the abstract.

We added this information to the abstract.

Also, mention of MHW-OAX events are 1.8 in 100 months does not indicate this is for monthly gridded products as described in lines 96 -104 of the results section. The first part of the abstract has extreme events in months per 100 months (using monthly gridded data products) and the last part of the abstract changes to events in days per year. It would be helpful to the reader to identify that daily model output was used for the latter results.

We now write “Daily model output from a large-ensemble simulation of an Earth system model is analyzed to assess changes in MHW-OAX likelihood under climate change. These changes in MHW-OAX likelihood arise due to mean warming and acidification, due to changes in variability of temperature and [H⁺], and due to changes in their interdependence.”

To avoid redundancy, the reference to the model simulation is now removed from line 11 where we now write: “Using monthly open-ocean observations, we show that globally 1.8 in 100 months (or one out of five present-day MHW months) are compound MHW-OAX event months, almost twice as many as expected for 90th percentile extreme event exceedances if MHWs and OAX events were statistically independent.”

I would also add to line 24 the increase to 265 days under 2C global warming is relative to a fixed pre-industrial baseline.

We now write “Among these changes, it is the mean warming and acidification that has the largest effect on the number of MHW-OAX days per year, increasing it from 12 to 265 days per year at 2°C global warming relative to a fixed pre-industrial baseline.”

The sentence in line 25 mentioning the 60% increase in events for a shifting mean baseline may also benefit by including something like "...a 60% increase (12 to 19 days)....". This is to reduce confusion around the shift from months per 100 months, to days per year, to percent change - all of which occur in one long paragraph.

We now write: “Even when mean trends are removed, an increase in [H⁺] variability leads to a 60% increase in the number of compound MHW-OAX days per year (from 12 to 19) under 2°C global warming.”

Line 51. The paper referenced as number 21 is listed in as submitted to Nature in 2020. Is it published?

Yes, it is published now. We updated the reference to <https://www.nature.com/articles/s41586-021-03981-7>.

Line 86. Closed off bracket is missing at end of sentence on this line. The authors should also change "daily data" in this line to "daily model output". The daily results are modelled not observed.

Changed as suggested.

Line 113. The [H⁺] in the eight time series sites are not all directly measured as far as I am aware. SST is directly measured and [H⁺] is often calculated from carbon system data. Changing from "...eight directly measured SST and [H⁺]...", to "...eight SST and [H⁺]..." would fix this statement.

We thank the reviewer for the comment. We picked those eight stations, because [H⁺] was there either measured itself or could be calculated based on measured T, SST, C_T, and A_T. For the remaining autonomous buoy stations, A_T needed to be estimated using LIARv2 and observations of SST and S. We adapted the text as proposed by the reviewer.

Lines 152-206. The authors provide a great description on the LMF pattern change. One issue for the Arctic and Southern Ocean is there are so few data available, and the models indicate considerable regional variability (Fig 4). The potential to disrupt ecosystems is compounded by these waters being close to undersaturation, so crossing to undersaturated waters may be more significant for the ecosystems of these regions than the co-occurrence of extreme events described in the paper. here is no mention of this in the paper.

As suggested by the reviewer, we now emphasize the importance of absolute thresholds for saturation states:

“Thus, on global average only 2% of months with high SST-[H⁺] compound events are also months with extremely low aragonite saturation state. That MHW-OAX events are usually not accompanied by extremely low calcium carbonate saturation state may prevent calcifying organisms from additional stress due to impacts on calcification and shell dissolution (Kroeker et al., 2013). However, the long-term declining trend in aragonite saturation state that is expected for the 21st century likely results in more frequent compound extremes in temperature and aragonite saturation state relative to fixed baselines and to more frequent occurrence of co-occurring high temperatures and aragonite undersaturation (where aragonite saturation state is below one) in polar and coastal oceans.”

Line 209. The paragraph starting here does describe the different scenarios for extreme events, but would it not be simpler for the reader to state at the beginning what this section is about i.e. "The frequency of MHW-OAX days is projected to change with climate (Fig. 4) and the capacity for marine organisms to adapt to the extreme events will vary. Changes in MHW-OAX occurrence will arise from an increase in the mean state of temperature and [H⁺], from changes in the variability of temperature..."? As with other sections of this paper I find the reason for a section is often further down the page. This is a different writing style and may not be the preference of the authors, but it may make reading the paper easier.

Changed as suggested by the reviewer:

“The occurrence of MHW-OAX days is projected to change with climate change (Fig. 4) and the capacity for marine organisms to adapt to the extreme events will vary.”

Line 212-222. The sentences in these lines would be useful at the end of the introduction. It describes why different methods for calculating LMF were used (fixed baseline versus trend removal), but this is again another decision for the authors. For me, it makes more sense.

We decided not to move these sentences to the end of the introduction, where the likelihood multiplication factor is discussed. A discussion of the different baselines for the changes in compound events at that point might be confusing, given that the first part of the results section discusses the compound event likelihood and likelihood multiplication factor for the present-day under a fixed present-day baseline.

Line 243. In Figure 4 caption, a dashed line is mentioned, and I cannot see this apart from a small squiggle in 4b (blue dashes above the solid blue line at just over 1C warming). Is it relevant?

The dashed line is barely visible, because the global change in (compound) extremes mainly depends on the global warming level rather than the warming path. The small squiggles in Figure 4b (but also in 4a and 4c)

arise during the second part of the 21st century under RCP2.6. We decided to show these dashed lines to demonstrate that the results are pathway independent. We modified the caption to improve clarity:

“The dashed lines in (a-c) show ensemble mean changes relative to warming levels under the RCP2.6 scenario. Differences between the RCP8.5 and RCP2.6 greenhouse gas scenarios are barely visible because the scenario spread is much smaller than the ensemble spread, indicating that the projected changes in MHW (Frölicher et al., 2018), OAX, and compound MHW-OAX event days under global warming are independent of the warming path.”

Line 253-, For Figure 4a a cause of the rapid rise in OAX events is mentioned in Line 259-260. I appreciate the authors are focused on OAX-MHW events. The figure 4a is the first thing that will be looked at by a reader and why the reason for the rapid increase in OAX events is not made apparent first up will confuse some readers. The following is just personal preference and if the authors do not agree, I have no issue with this. However, I find I am continually stopping reading to cross check the figures and text it takes a long time to get through the paper and this is frustrating.

"For a fixed preindustrial baseline, [H⁺] reaches a near-permanent extreme state of more than 360 days per year (Fig. 4a) as the seasonal variability in H⁺ exceeds the preindustrial values at global warming of 0.3C, or when atmospheric CO₂ exceeds 340ppm. Here, the increase in compound MHW-OAX events is mainly determined by the increase in MHWs as the near-permanent H⁺ extreme state causes, by definition, all the MHWs to be MHW-OAX events. For the preindustrial baseline, the occurrence of MHW-OAX events in Figure 4a is simulated to have increased 14-fold from 12 days per year on average at preindustrial to 167 days per year (165-169, 90% confidence interval) at 1°C global warming. Under continued global warming, MHW-OAX occurrence is projected to increase to 265 (263-266; 22-fold increase) days per year for 2°C warming and to 307 (307-308; 26-fold increase) days per year for 3 °C."

We agree that this paragraph (lines 253-270) was not structured ideally, first discussing MHW-OAX events, then discussing MHWs and OAX events individually, and then discussing MHW-OAX events again. We therefore rewrote the paragraph to:

“Relative to a fixed preindustrial baseline (see Methods), [H⁺] reaches a near-permanent extreme event state of more than 360 days per year already at a global warming of 0.3 °C when atmospheric CO₂ exceeds 340 ppm (Fig. 4a), due to the large increase in mean [H⁺] compared to the natural variability in [H⁺] anomalies. A near-permanent [H⁺] extreme state causes, by definition, all MHW days to be also MHW-OAX event days (Fig. 4a). As a result, the increase in compound MHW-OAX event days per year is mainly determined by the increase in MHWs and therefore long-term ocean warming (Frölicher et al., 2018; Oliver et al., 2019). The occurrence of MHW-OAX event days per year is simulated to have increased 14-fold at 1°C global warming, from 12 days per year on average at preindustrial to 167 days per year (165-169, 90% confidence interval) (Fig. 4a). Under continued global warming, MHW-OAX occurrence is projected to increase to 265 (263-266; 22-fold increase) days per year for 2 °C warming and to 307 (307-308; 26-fold increase) days per year for 3 °C. The largest increases in MHW-OAX event days per year are projected in the tropical regions of the Atlantic, the western Pacific, and the Indian Ocean (Fig. 4d). There, increasing temperatures exceed the relatively small natural variability earlier than in most other places (Frölicher et al., 2016) and thus lead to relatively larger increases in MHW-OAX event days per year and even near-permanent MHW-OAX events (hatched area in Fig. 4d). These permanent MHW-OAX events are projected to occur in 42 (42-43)% of the ocean surface area under 3 °C warming but can be largely avoided under 2 °C warming (10 (9-10)%). In regions, where SST is projected to decrease over the 21st century, such as in the North Atlantic south of

Greenland (Winton et al., 2013; Gervais et al., 2018) and parts of the Southern Ocean (Manabe et al., 1991; Haumann et al., 2020), the likelihood of MHWs and hence compound MHW-OAX event days per year decreases (Fig. 4d). This decrease occurs despite the co-occurring transition to near-permanent OAX events, indicating MHW-OAX occurrence is more strongly reduced by the decreases in MHW days than it is increased by the increases in OAX days."

Lines 267-270. Is there a reference to the projected decrease in SST for waters of the Southern Ocean and near the Greenland Sea and why this would occur under global warming over the 21st century (stratification change etc)?

For the North Atlantic, it has been suggested that the slowdown of the AMOC leads to a smaller northward heat transport and hence a reduced warming or cooling of the North Atlantic surface waters over the 21st century. We added the references Winton et al. 2013 (<https://doi.org/10.1175/JCLI-D-12-00296.1>) and Gervais et al. 2018 (<https://doi.org/10.1175/JCLI-D-17-0635.1>).

For parts of the Southern Ocean, such as in the Weddell Sea and Ross Sea, the transient surface ocean temperature change is expected to be small due to the strong coupling of surface waters to the cold deep ocean (Manabe et al., 1991). The simulated temperature decrease may be connected to a cooling of surface waters due to an increase in stratification partly associated with increased sea-ice-induced freshwater fluxes, resulting in less upward heat flux from warmer subsurface waters (Haumann et al., 2020).

The references Manabe et al., 1991 ([https://doi.org/10.1175/1520-0442\(1991\)004<0785:TROACO>2.0.CO;2](https://doi.org/10.1175/1520-0442(1991)004<0785:TROACO>2.0.CO;2)) and Haumann et al., 2020 (<https://doi.org/10.1029/2019AV000132>) have been added for the Southern Ocean.

Line 576. Mocsy 2.0 is not the standard used to calculate the marine CO₂ system. It may be for modellers, but not observationalists. This claim does not need to be included.

The claim was deleted as suggested by the reviewer. The sentence reads now: "If calculated pH was not provided by the data set, mocsy 2.0 (Orr et al., 2015) was used to calculate [H⁺] when C_T was measured and CO₂SYN (van Heuven et al., 2011) was used when pCO₂ was measured."

Line 626 - 633. The paper discusses discrepancies between LMF estimates for the Hawaii time series station (HOTS). Are the authors stating the LMF values for the monthly gridded values do not agree with the monthly HOTS carbon chemistry sampling, but do agree when the 3-hourly pCO₂ data is used to calculate monthly mean values? This is surprising given the HOTS site is in oligotrophic waters and the gridded data products are heavily constrained in the region by the WHOTS data and ship-based pCO₂ data. It is probably one of the best sampled sites in the ocean. Perhaps it indicates the calculation still has surprises, particularly given the limited number of time series sites.

We also think that the disagreement at HOT is somehow unexpected. Several potential reasons for the mismatch exist, such as (1) that the local conditions at the HOT site are not well represented by the averaged time series over close grid cells in the gridded data product, (2) that too much interpolation in the sparse gridded data product, such as from the neural-network based interpolation algorithm by

Landschützer et al., causes it to fail in reproducing the local conditions at HOT, or (3) that the difference between the point measurements at the HOT site and the monthly-mean data of the data product causes the disagreement. In addition, a large part of the disagreement may be due to the statistical uncertainty in the HOT estimate.

Overall, this is an important question raised by the reviewer and should be investigated in more depth in subsequent studies. No changes are made to the manuscript.

How is the LMF data accessed?

We are not sure if we correctly understood this question. In general, LMF is calculated as the ratio between the percentage of time steps where $[H^+]$ and temperature are both above the respective thresholds divided by the percentage of time steps where $[H^+]$ is above the threshold times the percentage where temperature is above the threshold (Eq. 1). At the HOT station, for example, $[H^+]$ is above its threshold in 30 out of 299 measurements, temperature is above its threshold in 30 out of 299 measurements, and $[H^+]$ and temperature are simultaneously above their respective thresholds in 3 out of 299 measurements. Therefore the LMF is $3/299/(30/299)^2=1.0$. We hope to have clarified the LMF calculation with this example.

I strongly recommend the authors add a sentence or two in the conclusions about the need to expand the observation base to better characterize change and the related biological impacts. Otherwise, the paper just ends with a "here is what we found statement", whereas the value will increase by highlighting the need to expand to coastal and under-sampled regions, particularly if the model predictions of change are correct.

We added the following to the final paragraph of the conclusions:

“Finally, this study also highlights the need for carbonate system observations on high temporal and spatial resolution to assess and quantify biogeochemical compound events, particularly in coastal and under-sampled high-latitude regions.”

Response to comments by Referee 5

This article has been reviewed by other reviewers, however this is my first reading of the manuscript. This paper examines the co-occurrence of marine heatwave and high $[H^+]$ events using a combination of gridded observations, a large ensemble climate model and selected station data. The authors show that compound events are more likely than randomly distributed compound events would be across most of the ocean (except in the tropical Pacific and high latitudes) and examine the causes for this.

This is a new and significant area of research, given the likelihood that compound stressors will produce synergistically negative effects in marine organisms. In general, this is an interesting article with some interesting and useful insights. I do however have a few concerns/questions detailed below.

We thank the reviewer for this positive assessment and the capture of several small editorial errors that otherwise likely would have slipped through.

General comments:

I find the reference to terms like events/frequency somewhat misleading. I suspect that most readers would use an event to refer to a continuous event with extreme conditions that can last for extended periods of time and the frequency to refer to the number such events per time period. Here an event refers to each

month above the extreme threshold and frequency refers to the number of months (rather than the number of events). I feel a clearer terminology is needed

We agree that frequency is often used to express how many distinct events (each potentially last many time steps) occur within a time period (e.g., Oliver et al., 2020). Therefore, we modified the text throughout the manuscript such that frequency of event is no longer used as a synonym for the metric ‘compound event months / days per year’. Instead, we now use the term likelihood that describes the probability that a given month or day is under extreme event conditions. For example, we now write in line 22:

“Among these changes, it is the mean warming and acidification that has the largest effect on the number of MHW-OAX days per year, increasing it from 12 to 265 days per year at 2°C global warming relative to a fixed pre-industrial baseline.”

In line 74, we now write:

“We here quantify the number of MHW-OAX months per year (for the observation-based data) and the number of MHW-OAX days per year (daily-mean model output), irrespective of whether these months or days belong to the same ongoing MHW-OAX event. Similarly, we define the likelihood multiplication factor (LMF) that quantifies the likelihood that a month or day is under MHW-OAX conditions relative to the expected likelihood if MHWs and OAX events would occur independently from each other. The LMF is the ratio between the observed likelihood of compound event months or days $p(\text{MHW} - \text{OAX day or month})$, calculated as the percentage of months or days that are under MHW-OAX conditions over a given period, and the theoretical likelihood of compound event months or days if SST and $[\text{H}^+]$ were statistically independent. The theoretical likelihood of compound event months or days for statistically independent variables is given by the product of the individual likelihoods of MHWs and OAX months or days $p(\text{MHW day or month}) \times p(\text{OAX day or month})$ (Zscheischler et al., 2017):”

Its not clear to me that the gridded datasets based only on ship data would actually be able to capture extreme events, given that these datasets are based on sparse data. If for example you do a scatter plot of station based vs gridded SSTA or $[\text{H}^+]$ do you actually get a reasonable match?

While the gridded observation-based product based on the available observations is not ideal to capture extreme events, it is the best available option now. When compared to the station-based data as suggested by the reviewer, we see that the station-based observations and the gridded data-based product match reasonably well (see Figure below). In general, the gridded data-based product can capture the SST variability but seems to underestimate the $[\text{H}^+]$ variability. For example, the station data for $[\text{H}^+]$ at CARIACO is much more variable, in this case likely because of the sites’ location near the coast.

R1: Scatter plots for $[H^+]$ and SST anomalies. ‘station’ refers to the time series (Ext. Data Table 2) and ‘gridded data’ refers to the gridded data product, where the time series were averaged over the 9 closest grid cells to the time series site (see Ext. Data Table 1).

However, the likelihood of an MHW-OAX event solely depends on how $[H^+]$ and SST co-vary with respect to each other and not on the magnitude of $[H^+]$ and SST variations. To capture the LMF, it is thus important whether the station data confirms the large-scale variations in dependence between SST and $[H^+]$. Therefore, we quantified the Pearson correlation coefficient

Corr. Coef.	KNOTK2	HOT	BATS	ESTOC	137E30N	137E5N	Munida	CARIACO
Station	-0.57	0.33	0.44	0.81	0.54	0.33	0.12	0.37
Gridded	-0.11	0.63	0.52	0.41	0.56	0.22	-0.21	0.32

The relatively good agreement in the correlation coefficient and the LMF (Ext. Data Table 1) suggests that the gridded data can be used to constrain the large-scale patterns of present day LMF. However, the comparison also highlights that the gridded product might underestimate the intensity of OAX events during MHW-OAX events.

I find the biological argument for a moving baseline rather unconvincing. This scenario assumes that the species is able to adapt to large changes in mean climate, but unable to adapt to much smaller changes in variability. While it is certainly interesting to separate changes in the mean from the variability to understand the physical processes at play, I'm not sure what biological relevance this really has.

The fixed baseline can indeed be expected to be overall biologically most relevant. Therefore, we now write in line 217: “This baseline is chosen to show the overall changes in MHW-OAX occurrence and because the fixed baseline is expected to be the most relevant baseline definition in many cases, in particular when projecting impacts for less resilient and less mobile organisms such as warm water corals (Hughes et al., 2017; Prada et al., 2017) or other sessile organisms that cannot adapt fast enough to long-term ocean warming and acidification or cannot relocate to favorable ocean habitats (Oliver et al., 2021).”

Nevertheless, Holbrook et al. (2020), Burger et al. (2020), Oliver et al. (2021), and Gruber et al. (2021) also highlight and discuss the biological importance of the shifting-mean baseline approach, pointing out that the ability to cope with the long-term trends does not imply that an organism is also able to cope with more frequent large departures from mean conditions during the extremes relative to a shifting mean baseline, because changes there occur on timescales of days to weeks, compared to the decadal time scales of the long-term trends.

We now write (see also below):

“Under the shifting-mean baseline, long-term warming and acidification trends are removed. Hence, extremes are defined as extreme deviations from the mean conditions that themselves change over time (Oliver et al., 2021; Burger et al., 2020; Jacox et al., 2019; Holbrook et al., 2020; Gruber et al., 2021). Changes in compound MHW-OAX event occurrence are here mainly caused by changes in temperature and especially [H⁺] variability (Oliver et al., 2021; Burger et al., 2020). This baseline is chosen to analyze the role of changes in variability and to quantify the stress for organisms due to increases in extreme deviations from the mean conditions. It is most meaningful for organisms that may adapt to the long-term warming and acidification trends (Munoz et al., 2015) or shift their distribution in response to these trends (Oliver et al., 2020; Pinsky et al., 2013; Cheung et al., 2021), such as fishes or marine mammals, because these organisms may still be affected by extreme deviations on much shorter timescales during individual events.”

Holbrook, N.J., Sen Gupta, A., Oliver, E.C.J. et al. Keeping pace with marine heatwaves. *Nat Rev Earth Environ* 1, 482–493 (2020). <https://doi.org/10.1038/s43017-020-0068-4>

Eric C.J. Oliver, Jessica A. Benthuyzen, Sofia Darmaraki, Markus G. Donat, Alistair J. Hobday, Neil J. Holbrook, Robert W. Schlegel, Alex Sen Gupta. *Annual Review of Marine Science* 2021 13:1, 313-342

Gruber, N., Boyd, P.W., Frölicher, T.L. et al. Biogeochemical extremes and compound events in the ocean. *Nature* 600, 395–407 (2021). <https://doi.org/10.1038/s41586-021-03981-7>

As you show in Fig 4a a small amount of warming means that you very quickly reach permanent H⁺ extreme conditions (presumably because H⁺ variability is so small compared to the climate change signal). This would suggest to me that the 90th percentile is not a very useful metric. Indeed if 0oC global warming represents pre industrial conditions, then we should have reached permanent H⁺ extreme by the mid to late 20th century. I think this point at least requires some discussion.

Yes, [H⁺] extremes relative to a fixed baseline increase quickly in frequency due to the increases in surface [H⁺] due to the oceanic uptake of anthropogenic carbon. As a result [H⁺] extremes under this definition become near permanent at around 340 ppm atm. CO₂ or at 0.3°C global warming.

The biological implication of permanent [H⁺] state is not straight-forward, because species-specific relations between relative thresholds and biological thresholds are not available. This is a general drawback when defining extreme events based on relative thresholds and not absolute thresholds. As this study provides a first general characterization of MHW-OAX events, their drivers, and their changes on the global scale, it relies on a first characterization of different aspects of changes in MHW-OAX events (e.g., under the different baselines). We hope that our study introduces useful concepts for future research efforts that identify species- and region-specific impacts on marine organisms under increasing occurrence of MHW-OAX events. We now discuss this aspect in the conclusion section:

“The biological impacts of these changes in MHW-OAX events across different species and ecosystems are currently largely unknown (Gruber et al., 2021). The potential threat from rising numbers of MHW-OAX days highlights the urgent need to better understand the organism and ecosystem response to such ocean compound events. Future studies on extreme events should also carefully choose the baseline depending on the impact which they analyze and potentially use absolute thresholds for specific species. Choosing the wrong baseline, shifting-mean for unmovable corals, or fixed for fish that can migrate, may lead to an overestimation or underestimation of the impact of changes in extreme events. Finally, this study also highlights the need for carbonate system observations on high temporal and spatial resolution to assess and quantify biogeochemical compound events, particularly in coastal and under-sampled high-latitude regions.”

Specific comments:

98: larger than one OVER 65% of the ocean

Changed as suggested.

Fig 1 for KNOT and K2 station the SSTA-[H⁺] correlation is negative, however the associated circle shows a positive LMF – how is this possible? The opposite seems to be the case for HOT i.e. positive correlation but -ve LMF.

This apparent mismatch may be a consequence of the limited number of observations. When the number of observations is as small as for the time series (63 to 457), the resulting LMF has large uncertainties

(Extended Data Table 1). For the HOT station, for example, the correlation coefficient indicates an LMF of 2.3. However, the uncertainty range goes from 0.3 to 2.4 due to the low number of measurements and encompasses this LMF estimate of 2.3 that is derived from the correlation coefficient.

Furthermore, the tail dependence of $[H^+]$ and SST, determining how often extremes in $[H^+]$ and SST co-occur, can be generally different from the correlation coefficient of their overall distributions. This aspect may add to the apparent miss-match between correlation and LMF.

No changes are made to the manuscript.

120: It is hence approximated by the Pearson correlation coefficient ...

I think this terminology is confusing. The sentence makes it sound like the correlation coefficient is approximately equal to the LMF. I don't believe this is what you mean, you mean that the two are related.

We agree and now write:

“However, the LMF can be estimated from the Pearson correlation coefficient (in the following simply correlation coefficient) of SST and $[H^+]$ anomalies (Ext. Data Fig. 1; see Methods), which can be mathematically decomposed.”.

126: The freshwater CYCLING term quantifies the direct impact

Changed as suggested.

127: CT and AT ...

AT hasn't been defined yet

We now define it in the previous sentence, writing “..., as well as smaller contributions from variations in salinity-normalized alkalinity (sA_T) and...”.

156: Positive anomalies in SST, such as during MHWs, are often connected to negative anomalies in sCT as these processes often lead to opposite changes in temperature and sCT

This sentence seems circular to me?

“These” referred to the physical and biogeochemical processes listed in the sentences before. To avoid ambiguity, we now write “Positive anomalies in SST, such as during MHWs, are often connected to negative anomalies in sCT as the aforementioned physical and biogeochemical processes often lead to opposite changes in temperature and sCT .”.

158: and enhanced thermal stratification

During El Nino the thermocline becomes deeper in the east, which means its harder for the deep waters to be brought to the surface the surface – I don't think this is manifest as a greater stratification.

Thanks for the pointer. We write now: “For example, weaker surface winds and a deepening thermocline in the central and eastern tropical Pacific during El Niño conditions drive high sea surface temperatures

(Holbrook et al., 2019; Oliver et al., 2021; Sen Gupta et al., 2020) but at the same time low s_{CT} due to reductions in mixing and upwelling of colder s_{CT} -rich waters.”

164-165: Here is where I would refer to stratification. At mid latitudes MHWs are associated with enhanced stratification that would suppress mixing of nutrients between surface and deep layers

We now write: “In warm, nutrient-poor regions (40°S–10°S; 10°N–40 °N), high temperature anomalies may reduce the nutrient supply from mixing due to an increase in thermal stratification”

189: that the reduction in s_{CT} and hence $[H^+]$ due to suppressed upwelling and mixing during MHWs ... It will also be related to the deeper pycnocline

To be consistent with the changes that we made above, we again refer to the thermocline instead of the pycnocline. We now write: “Thus, the reduction in s_{CT} and hence $[H^+]$ during MHWs in this region due to suppressed upwelling that coincides with a deepening thermocline must be large enough to overcompensate the positive temperature and biology contributions to $[H^+]$ and ultimately result in an LMF below 1.”

219: organisms that cannot adapt fast enough to ...

Your assumption here is not that they cant adapt fast enough, its that there is no adaptation (you are fixing the baseline).

We are not sure about the question. We think that the rate at which organisms can either adapt to the changing conditions or at which they can relocate is critical. While there is evidence for adaptive potential on decadal timescales, this potential is species dependent (e.g., Sunday et al., 2011; <https://doi.org/10.1371/journal.pone.0022881>), and organisms that can only adapt on longer time scales are likely impacted by warming and acidification trends and changes in MHW-OAX frequency relative to a fixed baseline.

225: that may ADAPT to long-term ...

Changed as suggested.

225 (and also in the discussion): or can shift their distribution ...

This doesn't make sense to me. Why would a moving baseline be appropriate for mobile species? If an organism can adapt, then it wouldn't need to move - poleward migration occurs because an organism cant adapt to warming temperature.

The idea here is here that organisms can cope with mean warming and acidification by either adapting to the changing conditions or by shifting the distribution to another location where ocean temperature and acidity are more suitable for them. If an organism fails to adapt or relocate, it is most impacted by fixed-baseline extremes. However, organisms that have either adapted or relocated may still be impacted by increases in extreme departures from the changing mean conditions, as captured by a shifting-mean baseline. We extended the discussion:

“Under the shifting-mean baseline, long-term warming and acidification trends are removed. Hence, extremes are defined as extreme deviations from the mean conditions that themselves change over time (Oliver et al., 2021; Burger et al., 2020; Jacox et al., 2019; Holbrook et al., 2020; Gruber et al., 2021). Changes in compound MHW-OAX event occurrence are here mainly caused by changes in temperature and especially [H+] variability (Oliver et al., 2021; Burger et al., 2020). This baseline is chosen to analyze the role of changes in variability and to quantify the stress for organisms due to increases in extreme deviations from the mean conditions. It is most meaningful for organisms that may adapt to the long-term warming and acidification trends (Munoz et al., 2015) or shift their distribution in response to these trends (Oliver et al., 2020; Pinsky et al., 2013; Cheung et al., 2021), such as fishes or marine mammals, because these organisms may still be affected by extreme deviations on much shorter timescales during individual events.”

Fig 4, the curves appear to turn back on themselves. Is this because there is still some residual variability in the ensemble mean (i.e. temperatures don't increase monotonically)?

Yes, there is. Internal variability is not completely averaged out even with the 30-member ensemble. However, more importantly, external natural forcing such as volcanoes provide ensemble mean cooling but acidity extremes events still increase due to the ongoing increases in atmospheric CO₂. Volcanic cooling of the ensemble-mean atmospheric temperatures are still visible in the 21-year ensemble mean that is used in Figs. 4a-c:

R2: Ensemble-mean atmospheric temperature change over the historical period as simulated by the 30-member ensemble simulation by the ESM2M model. The dashed line displays annual averages, and the solid line shows a 21-year running mean. A second x-axis displays the atmospheric CO₂ concentration that forced the ensemble simulation.

269: the frequency of MHWs and hence compound MHW-OAX events decreases ...

It seems strange that in regions where sst shows little change (so MHW frequency shouldn't change much) but H⁺ extremes become a permanent feature, the result is a decrease in compound events.

It would indeed be strange if a small change in the MHW frequency and permanent $[H^+]$ extremes would lead to a decrease in compound events. However, the relatively small decrease in mean SST has a surprisingly large effect on MHWs. The regions with a decrease in compound events in figure 4d also exhibit about a 10-fold decrease in MHWs at 2°C global warming (figure below). If MHWs and OAX events in these regions were independent from each other at preindustrial times (which they aren't), a 10-fold increase in OAX events between preindustrial and 2°C global warming (to permanent OAX events), co-occurring with the 10-fold decrease in MHWs, would leave the number of MHW-OAX events unchanged. However, the simulated correlation coefficient in these regions at preindustrial is generally positive (Ext. Data Fig. 6b). When assuming the anomalies in SST and $[H^+]$ are well described by a bivariate normal distribution (Eqs. 2 and 3), and assuming a correlation coefficient of 0.5 for the anomalies at preindustrial times, one obtains a decrease in MHW-OAX days from 11.8 days per year at preindustrial to 3.65 days per year at 2°C global warming when OAX events become permanent and MHW days per year decline to 3.65, hence a reduction by 8.2 days per year. This example shows that MHW-OAX events can become rarer in these regions, despite the emergence of a permanent OAX extreme state, simply due to a sharp decline in MHWs.

We added more explanation in L269: “the number of MHW and compound MHW-OAX event days per year decreases (Fig. 4d). This decrease occurs despite the co-occurring transition to near-permanent OAX events, indicating MHW-OAX occurrence is more strongly reduced by the decreases in MHW days than it is increased by the increases in OAX days”

R3: Change in the number of MHW days per year at 2° of global warming relative to the pre-industrial period. Regions where the number of MHW days per year increases were masked out to highlight regions with decreasing count of MHWs.

275: 1.9-fold increase

It looks like there is a 3x increase in H^+ extreme days, and a small increase in MHW days, so why do we only get a 1.9x increase in compound events?

This point is similar the previous one. This line of thought would be only correct if OAX events and MHWs would occur independently from each other, which is not the case. When MHWs and OAX events tend to occur together (which they do on global average, since $LMF > 1$ globally), a tripling in OAX event days does not cause a tripling in MHW-OAX events. This statement becomes clearer when looking at an unrealistic extreme case: If MHWs and OAX events would always fall together, then a tripling in OAX days under constant MHW days would result in a constant number of MHW-OAX days, because the number of OAX days is bounded by the number of MHW days.

In addition, the increase in MHW-OAX events is reduced by the decreasing trend in these events due to the decreases in dependence of SST and $[H^+]$ (Fig. 4c, f). This aspect is mentioned in the text, where we write

“This indicates that the reductions in the dependence of SST and $[H^+]$ that reduce MHW-OAX event occurrence are overcompensated by an increase in $[H^+]$ and SST variability, resulting in a net increase in MHW-OAX event days per year relative to a shifting-mean baseline.”. No changes are made to the manuscript.

291: The reduction in correlation may be attributed to the over- proportional increase in the $[H^+]$ sensitivity with respect to CT in warmer, high CT waters

Is there a reason why this occurs?

We think this is a feature of the relatively complex carbonate chemistry: An increase in background C_T causes large increases in the sensitivity of $[H^+]$ with respect to variations in C_T . This sensitivity increase is connected to a decrease in the buffer capacity, i.e. reductions in the carbonate ion concentration. A perturbation in C_T (under constant A_T , i.e. caused by an increase in $[CO_2]_{aq}$) then causes a larger fluctuation in $[H^+]$, because less of the change in $[H^+]$ is buffered by the reaction $CO_3^{2-} + H^+ \rightarrow HCO_3^-$ (e.g., Egleston et al. 2010; <https://doi.org/10.1029/2008GB003407>). The increase in $[H^+]$ sensitivity to temperature with higher background C_T , on the other hand, is smaller. This increase in $[H^+]$ sensitivity to temperature is caused by a change in how variations in the dissociation constants through temperature variations cause variations in $[H^+]$.

No changes made to the manuscript.

334: anomalies FOR THE present-day

Changed as suggested.

335: ... albeit with a general positive bias

Changed as suggested.

376. Not sure what you mean by: allowed to localize

Changed to: “The combination of observations and models allowed to identify hotspots of MHW-OAX events,...”.

Methods:

I don't believe that the baseline periods have been defined for the observation (pre industrial is used for the model). MHW statistics are very sensitive to the baseline used.

Although MHWs are sensitive to the baseline or reference period, the current ordering of the Methods section introduces first the analysis methods and then the data (as proposed by reviewer 2 from the previous review iteration). The motivation for this ordering was that the ‘Observation-based data’ and ‘Model evaluation’ sections, that are part of the section on the data, depend on the analysis methods and should thus be located after the section on the analysis methods. We now add information about the present-day period in the section on analysis methods. We write in line 428: “The data for the present-day period (gridded

observation-based product over the baseline period 1982-2019 and time series data with varying observation periods (Ext. Data Table 2)) were linearly detrended prior to identifying the extreme events.”

We now also explicitly specify the baseline period in the caption of Fig. 1.:

“All data was linearly detrended prior to the analysis and the baseline period 1982-2019 was used to define the extreme events.”

406: For the time series data ...

You haven't introduced the timeseries or other datasets yet in the methods. I think it would make more sense to start the methods with the datasets used.

We see why it would be intuitive to first introduce the data. However, we decided to keep the current order that was suggested by reviewer 2 in the first review round. With that order, all analyses methods are introduced before the 'Observation-based data' and the 'Model evaluation' section, since these two sections refer to MHW-OAX events, LMF, the statistical tests, and the baselines definitions. These concepts should thus be introduced in the first part of the Methods section.

412: threshold is generally small

Perhaps I have misunderstood, but I don't understand how this effect can be small. If you are using a annually fixed threshold you would only pick out summer MHW for example.

You test this on the gridded data, but I wonder if taking a global average (which includes a mix of summer and winter from different hemispheres) hides much larger single hemisphere errors.

The explanation of the two different thresholds was not clear enough here. Both approaches account for the seasonal cycle.

In the first approach, the seasonally varying percentile threshold approach, the 90th percentile is calculated for each calendar month individually. The likelihood of a MHW is hence the same in December as it is in July. However, if the SST variability in December is larger than in July, the difference of the 90th percentile to the mean SST in December will be larger than in July. Hence, the same SST anomaly with respect to the mean seasonal cycle may be a MHW in December but not in July. Over the 38-year long observed period (1982-2019), this approach yields 38 data points for each calendar month to calculate the percentile thresholds.

In the second approach, the anomaly percentile threshold, all anomalies over the year are used to calculate the 90th percentile. In this approach the same deviation from the mean SST in July and December will cause a MHW. If the SST variability in December was larger than in July, there would be more MHWs in December than in January under this approach. Over the historically-observed period (1982-2019), this approach yields $38 \times 12 = 456$ data points to calculate the anomaly percentile threshold.

If the number of years in a time series is relatively small, as it is for the historical period, the anomaly percentile threshold provides 12 times more data points and is statistically more robust. Seasonally varying variability in SST may, on the other hand, introduce a seasonal bias under the anomaly percentile threshold but not under the seasonally varying percentile threshold approach. This second factor, however, appears to be relatively modest, such that also the spatial LMF patterns between both approaches are relatively similar.

R4: The LMF using seasonally varying percentile thresholds compared to the LMF using anomaly percentile thresholds.

We have slightly adjusted the text to improve clarity: “The difference between calculating the LMF using seasonally varying thresholds and using one annual percentile threshold for the monthly anomalies is generally small.”

437: The ensemble mean was smoothed with a 365-day running mean filter to remove its seasonal cycle. Doesnt this means that your analysis doesn’t separate between changes in extremes and changes in the seasonal cycle. Is this important, given that there are projected changes in SST seasonality?

We only remove the seasonal cycle from the ensemble mean when calculating the long-term trend in the 90th percentiles to obtain the shifting-mean baseline. This long-term trend is then added to the pre-industrial seasonally-varying thresholds. Under this baseline definition, changes in extremes can arise due the changes in sub-annual (i.e., daily variations with periodicities of less than one year), seasonal, and interannual-to-decadal variability. The shifting-mean baseline definition thus only excludes the long-term trends from changes in extremes. For SST, this means that changes in SST seasonality contribute to changes in MHWs relative to the shifting mean baseline. We note that the changes in MHW occurrence due to changes in sub-annual, seasonal, and interannual-to-decadal variability changes in SST are relatively small on a global scale, as can be seen from Fig. 4b.

440: as the 90th percentiles of the 30-value ensemble distributions for that day
How do you calculate a robust 90th percentile using only 30 values. Or do you use a window around each day of the year to increase your sample size.

It is correct that we calculate a 90th percentile from only 30 values, meaning that 3 values of the ensemble distribution are above that threshold at each day and 27 values below. As a result, day-to-day variability will to some extent imprint on these thresholds and on the compound event occurrence based on these thresholds. By reporting compound events per year, this day-to-day variability should average out to a good degree. Therefore, the sample size should be large enough. Moreover, this approach makes sure that the likelihood for OAX events and MHWs stays truly constant within the ensemble, allowing to track how changes in the dependence between SST and [H⁺] imprint on the likelihood that a day is under MHW-OAX conditions. A window approach to increase the sample size would be possible. However, the number of MHWs and OAX events would then inherently vary to some extent.

488: For those that read the methods first it would be useful to define CT (sCT) and AT (sAT) again here.

We agree and now write "...from salinity-normalized dissolved inorganic carbon (sC_T) and total alkalinity (sA_T), as well as...".

REVIEWERS' COMMENTS

Reviewer #5 (Remarks to the Author):

This is a re-review of Compound marine heatwaves and ocean acidity extremes
Burger et al

The authors have responded in detail to all my comments. And I think this is a well presented paper and will be a valuable contribution to the literature.

My only remaining concern lies in the applicability of the OAX threshold for telling us something useful about risk to organisms. I appreciate that the authors have already responded to a similar question, but I would suggest that we know that the threshold used here isn't particularly useful in terms of impacts. 0.3°C degrees of warming has already come and gone (I'm assuming that this is relative to pre-industrial, although I didn't see this explicitly stated in Fig 4) as such we should be in a permanent OAX in most regions now. But this doesn't seem to have led to major H+ related impacts. Indeed we are still in general more concerned about temperature extremes. I'm not suggesting a change to the analysis, but I feel some further discussion is warranted on a useful choice of H+ threshold.

Fig. 2 in correlation coefficient units;

But correlation coefficient is dimensionless. Perhaps it would be useful to point out that the sum of b & c is approximately equal to a) with a small residual over most of the ocean

156: is mainly determined by the ratios between the variabilities in SST and sCT anomalies and the variability in [H+] anomalies in the respective 157 region

How do you determine this? The standard deviation of SST, sCt and H+ variability have quite similar spatial patterns. Are you saying that when you calculate the relevant standard deviation ratios, the spatial variability is smaller than the spatial variability in the sensitivity terms (i.e. Fig S3b & e)? Perhaps it would be helpful to show the ratios i.e. σ_{SST}/σ_{H+} and σ_{sCt}/σ_{H+} to demonstrate your point.

169: poleward displacement of warm, low CT waters in the boundary currents ...

This is a strange way to put it: the boundary currents would drive changes in advection. You would normally talk about 'displacement' in the context of fronts (which can presumably also result in opposing changes in Ct and SST)

174: causing a reduction in chlorophyll (Fig. 3a) and possibly net primary production,
Isn't this back to front. A decrease in NPP would result in a decrease in chlorophyll

176: increasing temperatures are associated with higher chlorophyll
Would be worth explaining why this might be the case (e.g. refs 45,43)

208: In addition to circulation, ...

This should probably be a new paragraph

224: two different baselines

But below you talk about three baselines: fixed, shifting mean, fully adapting.

230 cannot adapt fast enough

I think I was unclear in my previous review. A fixed baseline assumes that the organisms cannot adapt at all. If they were able to adapt slowly you could apply a slowly moving baseline, somewhere between your two extreme baseline cases. i.e. I think you should remove 'fast enough'

238: or shift their distribution in response to these trends

This still does not make sense to me. If the organism moves it will go to a region with different variability anyway. This analysis relates to variability changes at fixed locations in space. So a moving baseline does not help us explain what might happen for moving species.

246 i.e., to identify the impact of dependence changes on changes in MHW-OAX event likelihood. Not very clear. Do you mean ie. To identify changes in compound events related to changes in processes that affect the sensitivity of H+ to SST and/or Ct

Fig 4a-c. Is the global warming relative to pre industrial?

268-272

0.3oC warming would have occurred some time in the 20thC. Given that we haven't seen devastating impacts globally as a result of acidification it would appear that the relevance for biology of the extreme acidification threshold used here is tenuous.

288: more strongly reduced by the decreases in MHW days than it is increased by the increases in OAX days

More than this, compound days are fully determined by MHW days and have no dependence on changes in OAX once saturation occurs

301: In most other regions, the ratio between increases under the shifting-mean baseline and the fixed baseline is even smaller.

It should be remembered that fixed baseline includes the effect of both changes in mean and changes in variability, so the ratio is not simply indicative of the variability effect/mean effect ratio

312: over-proportional

Not sure what you mean by over proportional. Perhaps: The reduction in correlation appears to relate to the fact that there is an overall increase in both [H+] and SST sensitivity with respect to CT, with the [H+] sensitivity dominating over most of the ocean

318 Regionally

Would be clearer to say: In some regions ...

393: likely results

...will likely result ...

398: allowed to identify

Allowed us to identify/ allowed the identification of

402: we find that one out of four MHWs are also compound MHW-OAX

Im a bit confused. If OAX reaches saturation at 0.3oC. And we have passed 0.3oC warming some time ago, wouldn't we expect every MHW to be a MHW-OAX event?

421: shifting-mean for unmovable corals

shifting-mean for unmovable corals that are unable to adapt ...

422: fixed for fish that can migrate

This analysis tells us about local changes. To be applicable to migrating species we would need to know where they would migrate to, so a moving baseline isn't useful. If they need to move to somewhere where the background state is similar, then a fixed baseline is still probably the most useful framework.

Regards

Alex Sen Gupta

Nature Communications manuscript NCOMMS-21-39497A
Compound marine heatwaves and ocean acidity extremes

Response to comments by the editor and reviewer #5

June 1, 2022

Summary:

We thank the editor and the reviewer for their positive evaluation of our manuscript and the additional helpful comments and suggestions. We revised the manuscript taking into account all comments by you and the reviewer.

In addition, some smaller errors in the analysis scripts were detected while going through all analyses once more. These concern the LMF estimates for the 137°E transect data (5°N and 30°N) and the LMF estimate for the ESTOC station. Furthermore, the color bar ticks in Supplementary Figure 3e were not labeled correctly. These errors are corrected in the revised manuscript version.

We hope that this revision finds your approval.

Friedrich Burger

Response to comments by Referee 5

This is a re-review of Compound marine heatwaves and ocean acidity extremes

The authors have responded in detail to all my comments. And I think this is a well presented paper and will be a valuable contribution to the literature.

Thank you!

My only remaining concern lies in the applicability of the OAX threshold for telling us something useful about risk to organisms. I appreciate that the authors have already responded to a similar question, but I would suggest that we know that the threshold used here isn't particularly useful in terms of impacts. 0.30C degrees of warming has already come and gone (Im assuming that this is relative to pre-industrial, although I didn't see this explicitly stated in Fig 4) as such we should be in a permanent OAX in most regions now. But this doesn't seem to have led to major H+ related impacts. Indeed we are still in general more concerned about temperature extremes. Im not suggesting a change to the analysis, but I feel some further discussion is warranted on a useful choice of H+ threshold.

Yes, Figure 4 shows indeed the warming relative to pre-industrial conditions. We have added this information to the figure caption.

We agree that in cases when a trend pushes a system toward near permanent extreme conditions, the use of a fixed pre-industrial baseline may not be the optimal choice. However, as this is the first global study on MHW-OAX event occurrence in the global ocean over time, we had to use an extreme event definition that is simple, widely used, and globally comparable. In the manuscript, we tried to highlight this simplification. More specific [H⁺] impact thresholds for specific groups of organisms and ecosystems would be preferable in a more detailed analysis but are unknown in most cases and would also not allow for a global analysis as conducted here. Please note that the permanent [H⁺] extremes are only occurring in the time evolution. When analyzing present-day observations (Figure 1, 2, and 3), we define extreme events with respect to present time, hence there is no permanent [H⁺] extreme event and this threshold may be more relevant for organisms and ecosystems.

We now also added more stress on the necessity of impact-focused studies on MHW-OAX events, adding to L417:

"The potential threat from rising numbers of MHW-OAX days highlights the urgent need to better understand the organism and ecosystem responses to such ocean compound events. In particular, the knowledge on the biological impacts of extreme conditions in [H⁺] is still limited. A way forward would be to identify biologically informed thresholds for SST and [H⁺] specific to key species of a certain region that directly relate such events to ecosystem impacts."

Fig. 2 in correlation coefficient units; But correlation coefficient is dimensionless. Perhaps it would be useful to point out that the sum of b & c is approximately equal to a) with a small residual over most of the ocean

Changed as proposed. We added: "The sum of panels b and c approximately equals to panel a."

156: is mainly determined by the ratios between the variabilities in SST and sCT anomalies and the variability in [H⁺] anomalies in the respective 157 region

How do you determine this? The standard deviation of SST, sCt and H+ variability have quite similar spatial patterns. Are you saying that when you calculate the relevant standard deviation ratios, the spatial variability is smaller than the spatial variability in the sensitivity terms (i.e. Fig S3b & e)? Perhaps it would be helpful to show the ratios i.e. σ_{SST}/σ_{H+} and σ_{sCt}/σ_{H+} to demonstrate your point.

Yes, the spatial patterns of the standard deviations are relatively similar (Ext. Data Fig. 3a, c, d). Nonetheless, the ratio between SST and [H⁺] standard deviation (Figure below, left side) and the ratio between sC_T and [H⁺] standard deviation (Figure below, right side) still have clear spatial patterns that mainly drive the spatial patterns of the SST and sC_T contributions to SST-[H⁺] correlation coefficient (Fig. 2 b-c). In contrast, the spatial patterns of the sensitivities (Ext. Data Fig. 3b, e) are more uniform and thus less important for the spatial patterns of the contributions to SST-[H⁺] correlation.

We decided not to show these ratios in the manuscript to avoid adding further figures.

169: poleward displacement of warm, low CT waters in the boundary currents ...

This is a strange way to put it: the boundary currents would drive changes in advection. You would normally talk about 'displacement' in the context of fronts (which can presumably also result in opposing changes in C_T and SST)

Changed to "poleward advection of warm, low C_T waters".

174: causing a reduction in chlorophyll (Fig. 3a) and possibly net primary production,

Isn't this back to front. A decrease in NPP would result in a decrease in chlorophyll

We modified the text to: "Causing a reduction in chlorophyll (Fig. 3), possibly co-occurring with a reduction in net primary production."

176: increasing temperatures are associated with higher chlorophyll

Would be worth explaining why this might be the case (e.g. refs 45,43)

We added: "increasing temperatures can stimulate phytoplankton growth and can reduce the light limitation through a shoaling of the mixed layer and possible increased shortwave radiation (Vogt et al., 2022) and are thus associated with higher chlorophyll and primary production".

208: In addition to circulation, ...

This should probably be a new paragraph

Changed as proposed.

224: two different baselines

But below you talk about three baselines: fixed, shifting mean, fully adapting.

Agreed. We now already mention the fully-adapting baseline upfront in the paragraph and refer to three baselines right away. Specifically, we write:

"To consider different adaptation capabilities of organisms and ecosystems, we define changes in the number of MHW-OAX days per year with respect to three different baselines (refs 7, 9, 21): relative to a fixed pre-industrial baseline, relative to a shifting mean baseline, and relative to a fully adapting baseline (see Methods)."

230 cannot adapt fast enough

I think I was unclear in my previous review. A fixed baseline assumes that the organisms cannot adapt at all. If they were able to adapt slowly you could apply a slowly moving baseline, somewhere between your two extreme baseline cases. i.e. I think you should remove 'fast enough'

Done.

238: or shift their distribution in response to these trends

This still does not make sense to me. If the organism moves it will go to a region with different variability anyway. This analysis relates to variability changes at fixed locations in space. So a moving baseline does not help us explain what might happen for moving species.

We agree that the argument was too coarse. We expand the discussion:

"It is most meaningful for organisms that may adapt to the long-term warming and acidification trends (ref. 53). Furthermore, it may also be more meaningful for mobile species, such as fishes or marine mammals, because these species may relocate along gradients in the mean conditions but may still

be impacted by more frequent variability-driven extremes (refs. 7,54,55), in particular if relocation is not possible on the short timescales of individual events.”

246 i.e., to identify the impact of dependence changes on changes in MHW-OAX event likelihood.
Not very clear. Do you mean ie. To identify changes in compound events related to changes in processes that affect the sensitivity of H+ to SST and/or Ct
Unequal changes in the sensitivities are one way of changing the statistical dependence between T and [H+]. We prefer to keep the sentence more general but clarify:
“i.e., to identify the impact of changes in statistical dependence between SST and [H+] anomalies on MHW-OAX event likelihood.”

Fig 4a-c. Is the global warming relative to pre industrial?

Yes. The information was added to the figure caption: “... relative to global warming levels with respect to pre-industrial conditions for MHWs (red lines),...”

268-272

0.3oC warming would have occurred some time in the 20thC. Given that we haven't seen devastating impacts globally as a result of acidification it would appear that the relevance for biology of the extreme acidification threshold used here is tenuous.

Please see our response at the beginning of the rebuttal letter.

288: more strongly reduced by the decreases in MHW days than it is increased by the increases in OAX days

More than this, compound days are fully determined by MHW days and have no dependence on changes in OAX once saturation occurs

Agreed. We rephrase “The decrease in MHW-OAX days occurs as a result of the strong decrease in MHW days and despite the co-occurring transition to near-permanent OAX events.”

301: In most other regions, the ratio between increases under the shifting-mean baseline and the fixed baseline is even smaller.

It should be remembered that fixed baseline includes the effect of both changes in mean and changes in variability, so the ratio is not simply indicative of the variability effect/mean effect ratio

To reflect the reviewers' we added:

“The overall much lower increase in MHW-OAX days under a shifting-mean baseline (caused by changes in variability) than under a fixed baseline (caused by changes in mean and in variability) reflects the dominant role of mean changes for the evolution of MHW-OAX events under a fixed baseline.”

312: over-proportional

Not sure what you mean by over proportional. Perhaps: The reduction in correlation appears to relate to the fact that there is an overall increase in both [H+] and SST sensitivity with respect to CT, with the [H+] sensitivity dominating over most of the ocean

We here refer to the over-proportional (i.e., relatively larger) increase in the [H+] sensitivity with respect to C_T compared to the [H+] sensitivity with respect to SST (see Ext. Data Fig 5).

We rephrase:

“The reduction in correlation may be attributed to the relatively larger increase in the [H+] sensitivity with respect to C_T than in the increase in [H+] sensitivity with respect to temperature in warmer, high C_T waters.”

318 Regionally

Would be clearer to say: In some regions ...

Changed.

393: likely results

...will likely result ...

Changed.

398: allowed to identify

Allowed us to identify/ allowed the identification of

Changed to “Allowed the identification of”.

402: we find that one out of four MHWs are also compound MHW-OAX

Im a bit confused. If OAX reaches saturation at 0.3oC. And we have passed 0.3oC warming some time ago, wouldn't we expect every MHW to be a MHW-OAX event?

As mentioned in the text, we use two reference periods throughout the manuscript. The 1982-2019 reference period for observational data and the pre-industrial reference period for model projections. Here, we refer to the 1982-2019 reference period that was used for the observation-based data. We thank for the pointer and clarify: "This suggests that some of the observed MHWs (ref. 6) were also compound MHW-OAX events, in particular in the low-to-mid latitudes where we find that one out of four MHWs are also compound MHW-OAX events when extremes are defined with respect to the 1982-2019 reference period."

421: shifting-mean for unmovable corals

shifting-mean for unmovable corals that are unable to adapt ...

Changed: "Choosing the wrong baseline, shifting-mean for unmovable corals that are unable to adapt to the long-term trends, or fixed for fish that can migrate along gradients in mean conditions, may lead to an underestimation or overestimation of the impact of changes in extreme events."

422: fixed for fish that can migrate

This analysis tells us about local changes. To be applicable to migrating species we would need to know where they would migrate to, so a moving baseline isn't useful. If they need to move to somewhere where the background state is similar, then a fixed baseline is still probably the most useful framework. The second part of the sentence starting in L421 was adapted by adding that we here consider a mobile species migrating along the mean gradients (see above).